# Ternatin and improved synthetic variants kill cancer cells by targeting the elongation factor-1A ternary complex

Jordan D Carelli[1], Steven G Sethofer[2], Geoffrey A Smith[1], Howard R Miller[3], Jillian L Simard[2], William C Merrick[4], Rishi K Jain[3], Nathan T Ross[3], Jack Taunton[2]*

[1]Chemistry and Chemical Biology Graduate Program, University of California, San Francisco, San Francisco, United States; [2]Department of Cellular and Molecular Pharmacology, University of California, San Francisco, San Francisco, United States; [3]Novartis Institutes for BioMedical Research, Cambridge, United States; [4]Department of Biochemistry, School of Medicine, Case Western Reserve University, Cleveland, United States

**Abstract** Cyclic peptide natural products have evolved to exploit diverse protein targets, many of which control essential cellular processes. Inspired by a series of cyclic peptides with partially elucidated structures, we designed synthetic variants of ternatin, a cytotoxic and anti-adipogenic natural product whose molecular mode of action was unknown. The new ternatin variants are cytotoxic toward cancer cells, with up to 500-fold greater potency than ternatin itself. Using a ternatin photo-affinity probe, we identify the translation elongation factor-1A ternary complex (eEF1A·GTP·aminoacyl-tRNA) as a specific target and demonstrate competitive binding by the unrelated natural products, didemnin and cytotrienin. Mutations in domain III of eEF1A prevent ternatin binding and confer resistance to its cytotoxic effects, implicating the adjacent hydrophobic surface as a functional hot spot for eEF1A modulation. We conclude that the eukaryotic elongation factor-1A and its ternary complex with GTP and aminoacyl-tRNA are common targets for the evolution of cytotoxic natural products.

*For correspondence: jack.taunton@ucsf.edu

**Competing interests:** The authors declare that no competing interests exist.

## Introduction

Macrocyclic natural products are a rich source of biologically active compounds, and identification of their targets can open up new opportunities for therapeutic intervention (*Driggers et al., 2008*; *Villar et al., 2014*). Cyclic peptide natural products, especially those containing one or more *N*-methyl (or *N*-alkyl) amino acids, comprise a privileged class of macrocycles due to their modular structure, relative ease of synthesis, intrinsic cell permeability, and constrained three-dimensional architectures (*Bockus et al., 2013*; *Giordanetto and Kihlberg, 2014*; *Hewitt et al., 2015*). Examples of cyclic peptides with well characterized cellular targets include cyclosporin (*Liu et al., 1991*), trapoxin and romidepsin (*Taunton et al., 1996*; *Nakajima et al., 1998*), and cotransin/CAM741 and decatransin (*Besemer et al., 2005*; *Garrison et al., 2005*; *Junne et al., 2015*). These macrocyclic compounds target conserved eukaryotic proteins that play essential roles in calcium-mediated signaling (cyclophilin/calcineurin), epigenetic regulation (histone deacetylases), and secretory protein biogenesis (Sec61 translocon), respectively. In addition to serving as drugs and potential leads for cancer and immune diseases, these cyclic peptides have proven to be useful as chemical probes, revealing new cellular roles for their target proteins, as well as mechanistic insights into the biochemical processes they orchestrate.

**eLife digest** Many plants, fungi, and bacteria have evolved to produce small molecules that have powerful effects on the cells of other living organisms, and can even kill them. These naturally produced compounds are often used as starting points for developing new drugs. One such class of compounds are the cyclic peptides, which can be relatively easily produced in the laboratory and are able to penetrate cells. Some cyclic peptides have also proved to be useful for treating cancer and immune diseases, so researchers are keen to identify others that have similar effects. One promising prospect, called ternatin, is produced by several species of fungi. In high doses, ternatin can kill mammalian cells, but it was not clear how it does so.

To learn more, Carelli et al. searched a chemical database for cyclic peptides related to ternatin and identified several similar compounds that were reported to kill cancer cells. Inspired by the structures of these cyclic peptides, Carelli et al. synthesized modified versions of ternatin. One of these was 500 times more potent than ternatin, which means a much lower dose of the compound is still able to kill cancer cells.

Further experiments showed that ternatin blocks the production of new proteins in cells. Specifically, ternatin binds to a complex that includes a protein called elongation factor-1A (eEF1A). Mutations in a particular region of eEF1A prevent ternatin from killing cells, suggesting a potential binding site for ternatin.

The next challenge is to dissect the mechanism by which compounds binding to this site on eEF1A block protein synthesis and kill cells. A related challenge is to understand why certain cancer cells are hypersensitive to ternatin and other eEF1A inhibitors, while other cancer cells are relatively resistant. These questions are relevant to the development of eEF1A inhibitors as cancer treatments.

Ternatin (**1**, *Figure 1a*) is an *N*-methylated cyclic heptapeptide that inhibits adipogenesis at low nanomolar concentrations ($EC_{50}$ 20 nM) and becomes cytotoxic at ten-fold higher concentrations (*Shimokawa et al., 2008a*). Previous work established an important role for leucine-4, as substitution with alanine (ternatin-4-Ala, **2**) abolished biological activity (*Shimokawa et al., 2008b*). Motivating the current study, ternatin's molecular target and mechanism of action were completely unknown. While searching for cyclic peptides that, like ternatin, incorporate a β-hydroxy leucine, we found a patent application that describes partially elucidated structures of five related cyclic heptapeptides isolated from an *Aspergillus* fungus in Malaysia (*Blunt et al., 2010*). All five compounds were reported to be cytotoxic to cancer cells in the picomolar to low nanomolar range ($IC_{50}$ 0.1–24 nM vs HCT116, MCF7, and P388 cells). One of the more potent congeners, termed 'A3', is shown in *Figure 1a*. Although only 4 out of 11 stereocenters in A3 were assigned, we were able to map the amino acid sequence, stereochemistry, and *N*-methylation pattern directly onto the structure of ternatin. The apparent structural similarity to ternatin was not mentioned in the patent application.

Based on the potential similarity between A3 and ternatin structures, we hypothesized that replacement of *N*-Me-Ala-6 and Leu-4 in ternatin with two A3-specific residues – pipecolic acid and dehydro-homoleucine, respectively – would be sufficient to confer increased cytotoxic potency. These considerations led to the design of cyclic peptides **3** and **4** as potential 'north/south' hybrids of ternatin and A3 (*Figure 1a*). Compound **3** differs from ternatin only at position 6, whereas **4** additionally has dehydro-homoleucine at position 4. Here, we demonstrate that ternatin and its structural variants inhibit cellular protein synthesis, with compound **4** showing the greatest potency. We further identify the elongation factor-1A ternary complex (eEF1A·GTP·aminoacyl-tRNA) as a direct target of ternatin-related cyclic peptides. This family of *N*-methylated cyclic heptapeptides represents a new structural class of macrocyclic elongation factor-1A inhibitors.

## Results and discussion

Synthesis of **3** and **4** proceeded uneventfully according to a route previously established for ternatin (see Materials and methods) (*Shimokawa et al., 2007*). We synthesized both C4 epimers of dehydro-homoleucine (2-amino-4-methylhex-5-enoic acid), maintaining the (*S*)-stereochemistry at C2 (by analogy to L-leucine at position 4 of ternatin) and varying the C4 side-chain stereochemistry; each

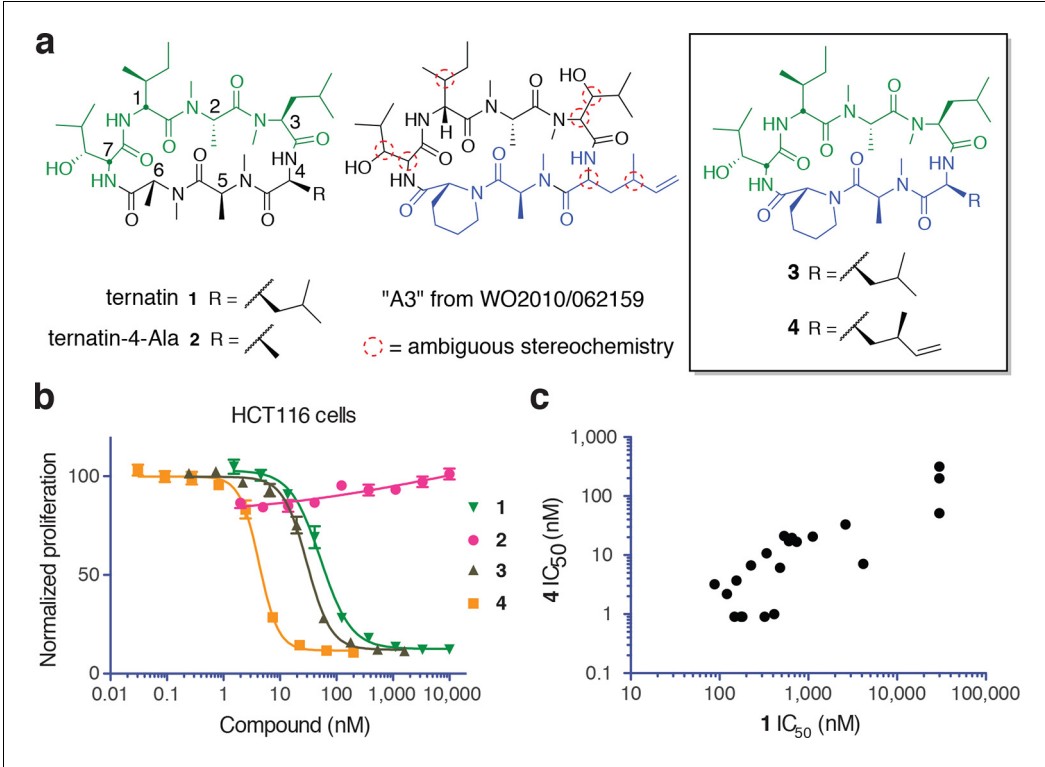

**Figure 1.** New ternatin variants inspired by *Aspergillus*-derived cyclic peptides. (**a**) Design of ternatin variants **3** and **4**, based on the partially elucidated structure of A3. (**b**) Effect of cyclic peptides **1**–**4**(three-fold dilutions) on HCT116 cell proliferation over 72 hr. (**c**) Compounds **1** and **4** were tested against a panel of 21 cancer cell lines. Shown is a scatter plot comparing $IC_{50}$ values of **1** and **4** for each cell line (Spearman correlation, 0.84, p < 0.0001).

The following source data and figure supplement are available for figure 1:

**Source data 1.** Antiproliferative activity of **1** and **4** against cancer cell lines.

**Figure supplement 1.** Epimers **4** and **4b** have similar antiproliferative activity.

---

epimer was incorporated separately into the final cyclic peptide. The (2*S*,4*R*)-epimer depicted in **4** (*Figure 1a*) was slightly more potent than the (2*S*,4*S*)-epimer in cell proliferation assays (*Figure 1—figure supplement 1*) and was used in all subsequent experiments.

We first tested the effects of compounds **1**–**4** on HCT116 cells. Ternatin (**1**) potently inhibited HCT116 cell proliferation ($IC_{50}$ 71 ± 10 nM, *Figure 1b*), whereas ternatin-4-Ala (**2**) had no effect up to 10 µM, consistent with its reported lack of activity in an adipogenesis assay (*Shimokawa et al., 2008b*). Ternatin-4-Ala (**2**) is thus a valuable negative control compound for mechanism-of-action studies. Structural modifications inspired by the *Aspergillus* cyclic peptides had significant effects on potency. While substitution with pipecolic acid increased potency two-fold, further substitution with (2*S*,4*R*)-dehydro-homoleucine resulted in an additional ten-fold increase ($IC_{50}$ 4.6 ± 1.0 nM). To extend these results, we tested **1** and **4** against a panel of 21 cell lines derived from diverse solid and hematological tumors. While $IC_{50}$ values of **1** and **4** spanned 3–4 orders of magnitude, they were correlated across all cell lines (Spearman correlation, 0.84, p<0.0001), with **4** being twenty-fold to >500-fold more potent than **1**, depending on the cell line (*Figure 1c*). This strong correlation suggests that ternatin and **4** kill cancer cells by a similar mechanism. Moreover, the pipecolic acid and dehydro-homoleucine substitutions in **4** resulted in dramatically increased potency across all cell lines tested.

Based on its broad anti-proliferative activity across many cell lines, we speculated that **4** might inhibit a fundamental cellular process. Indeed, global protein synthesis was abolished in cells treated with **4** (*Figure 2a*). Pulse labeling of newly synthesized proteins with [35]S-methionine revealed that **4**

inhibits protein synthesis more potently than ternatin, in agreement with the cell proliferation results. The inactive control, ternatin-4-Ala (**2**), had no effect on protein synthesis (*Figure 2b*).

Protein synthesis inhibitors tend to block either the initiation or the elongation phase of translation. To distinguish between effects on initiation and elongation, we analyzed ribosome/polysome profiles by sucrose density centrifugation after treating cells with **4**. The polysome profile was unchanged in cells treated with **4** compared to DMSO control (*Figure 2c*, left panel), indicating that global translation initiation was not perturbed. To test for inhibition of translation elongation, we asked whether **4** could abrogate polysome runoff induced by the initiation inhibitor harringtonine (*Lindqvist et al., 2010*). Similar to the known elongation inhibitor cycloheximide, pretreatment of cells with **4** completely blocked harringtonine-induced polysome depletion (2c, right panel). We conclude that **4** most likely inhibits translation elongation.

To identify the direct target of ternatins, we designed a photo-affinity probe. Previous efforts to prepare a biotinylated ternatin probe yielded an inactive analogue (*Shimokawa et al., 2009*). We therefore designed the bifunctional probe **5** (*Figure 3a*), which incorporates photo-leucine at position 4 and an alkyne at position 6 for tagging via click chemistry conjugation, similar to our previous strategy with cotransin cyclic peptides (*MacKinnon et al., 2007*; *MacKinnon and Taunton, 2009*). Probe **5** was active in a cell proliferation assay ($IC_{50}$ 460 ± 71 nM), albeit with ~ten-fold reduced

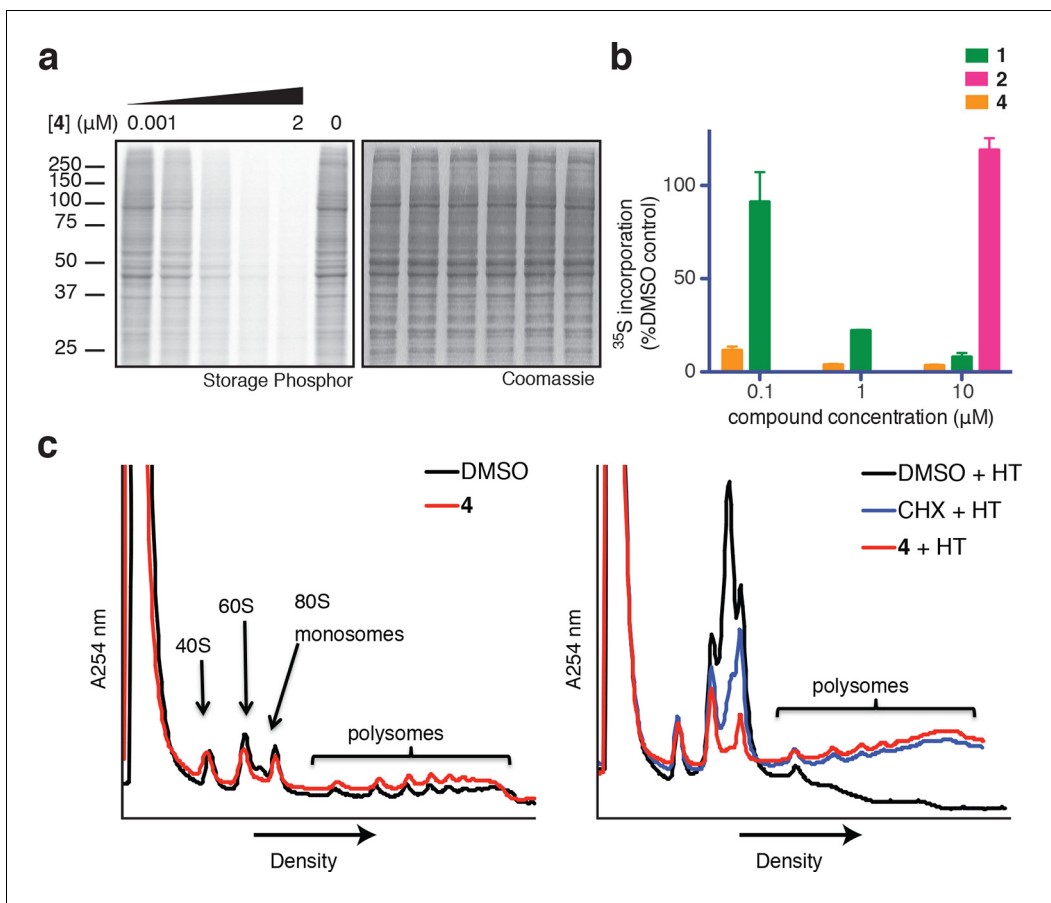

**Figure 2.** Ternatins inhibit global protein synthesis. (**a**) HCT116 cells were treated with compound **4** (five-fold dilutions) for 5 hr before labeling with $^{35}$S-Met for 1 hr. Cell lysates were separated by gel electrophoresis. Newly synthesized and total proteins were visualized by autoradiography and Coomassie staining. (**b**) Cells were treated as in (**a**), and $^{35}$S-labeled proteins were quantified by liquid scintillation counting after TCA precipitation (mean ± SEM, n = 3). (**c**) Left panel: HeLa cells were treated with **4** (5 µM) or DMSO for 20 min. Right panel: HeLa cells were treated with **4** (5 µM), cycloheximide (CHX, 100 µg/mL), or DMSO for 15 min, followed by harringtonine (HT, 2 µg/mL) for 20 min. After compound treatment, lysates were fractionated on 10–50% sucrose density gradients with absorbance detection at 254 nm.

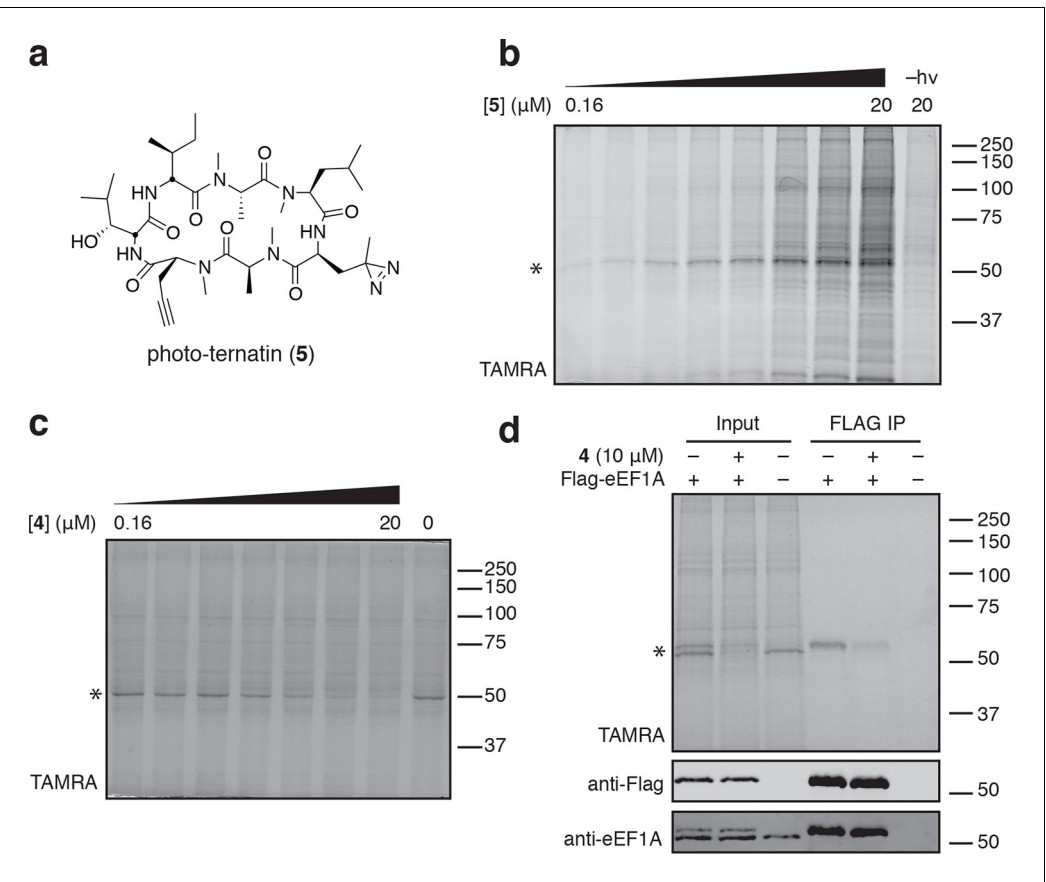

**Figure 3.** Photo-affinity labeling reveals eEF1A as a direct ternatin target. (a) Clickable photo-affinity probe **5**. (b) HEK293T cell lysates were treated with **5** (two-fold dilutions) for 20 min at room temp followed by UV irradiation (355 nm, 1000 W, 90 s). A control sample with 20 µM **5** was not irradiated. Samples were subjected to click chemistry with TAMRA-azide, separated by gel electrophoresis, and scanned for in-gel fluorescence. (c) Cell lysates were treated with increasing concentrations of **4** for 10 min before adding **5** (2 µM) for 20 min, UV irradiation, and processing as in (b). (d) HEK293T cells were transfected with Flag-eEF1A. Lysates were treated with probe **5** (2 µM) ± **4** and photolyzed as in (b), then immunoprecipitated with magnetic anti-Flag beads. Samples were eluted with SDS, subjected to click chemistry with TAMRA-azide, and analyzed by in-gel fluorescence scanning and Western blotting. Coomassie-stained gels corresponding to (b) and (c) are shown in *Figure 3—figure supplement 1*.

The following figure supplement is available for figure 3:

**Figure supplement 1.** Ternatin photo-affinity probe specifically labels a 50-kDa protein.

potency compared to ternatin (*Figure 3—figure supplement 1*). Treatment of cell lysates with increasing concentrations of **5**, followed by photolysis and copper-catalyzed click conjugation with tetramethylrhodamine (TAMRA)-azide, revealed preferential labeling of a 50-kDa protein (*Figure 3b*, *Figure 3—figure supplement 1*). Labeling of the major 50-kDa band, but not any of the minor bands, was prevented by pretreatment with competitor **4** (*Figure 3c*). These experiments demonstrate specific and saturable binding of ternatins to a 50-kDa protein in cell lysates.

Given that ternatins inhibit translation elongation, we hypothesized that the 50-kDa target protein was the eukaryotic elongation factor-1A (eEF1A). We tested this by photo-labeling lysates derived from cells expressing Flag-tagged eEF1A. Probe-labeled eEF1A was eluted from anti-Flag beads, visualized by subjecting the eluates to click chemistry with TAMRA-azide (*Figure 3d*). Photo-labeling of ectopically expressed Flag-eEF1A was reduced in the presence of excess **4** and was not detected in lysates derived from nontransfected cells.

eEF1A is an essential GTPase, orthologous to bacterial EF-Tu, whose canonical role is to deliver aminoacyl-tRNA (aa-tRNA) to a translating ribosome (*Andersen et al., 2003*). In cells, a significant fraction of eEF1A is thought to reside in a ternary complex bound to GTP and aa-tRNA (*Gromadski et al., 2007*). To test whether ternatins bind preferentially to an RNA-containing complex of eEF1A, we incubated cell lysates with RNase A, which can degrade most forms of RNA, including aa-tRNAs bound to eEF1A (*Tzivelekidis et al., 2011*). Adding RNase A to lysates before, but not after, photolysis completely prevented labeling of the 50-kDa protein by **5** (*Figure 4a*), suggesting that an RNA component is essential for photo-ternatin binding. Next, we tested whether **5** labels native eEF1A purified from rabbit reticulocytes. Strikingly, photo-crosslinking of **5** (1 μM) to purified eEF1A (1 μM) was observed only in the presence of GTP and aa-tRNA (in this case, Phe-tRNA; *Figure 4b*, lane 2). Photo-affinity labeling of the ternary complex was competed by excess **4** (10 μM). Addition of either GTP or GDP alone was not sufficient, and only weak nonspecific labeling was observed (lanes 1 and 4). These data indicate that ternatins preferentially target the ternary complex of eEF1A·GTP·aa-tRNA.

Didemnin B and cytotrienin A have also been implicated as eEF1A inhibitors, although these cytotoxic natural products are structurally unrelated to each other and ternatin. Didemnin (*Figure 4c*) has been reported to bind eEF1A·GTP with an affinity of 15 or 200 μM, depending on the assay conditions (*Ahuja et al., 2000*; *Crews et al., 1994*). While its mechanism of action is still debated (*Crews et al., 1996*; *Meng et al., 1998*; *Ahuja et al., 2000*; *Vera and Joullie, 2002*; *Potts et al., 2015*), didemnin has been shown to block an eEF1A-dependent step of translation elongation (*SirDeshpande and Toogood, 1995*; *Ahuja et al., 2000*; *Lindqvist et al., 2010*). Cytotrienin A, and likely the related polyketide ansatrienin B (*Figure 4c*), inhibit translation elongation by a mechanism similar to didemnin's (*Lindqvist et al., 2010*), although direct binding of these compounds to eEF1A has not been demonstrated. Using the ternary complex assembled from eEF1A, GTP, and Phe-tRNA, we found that didemnin and ansatrienin could compete with photo-ternatin **5** (*Figure 4d*). Similarly, didemnin and ansatrienin abolished photo-labeling of the 50-kDa band in cell lysates (*Figure 4—figure supplement 2*), further supporting the assignment of this band as eEF1A. These results suggest that didemnin, ansatrienin/cytotrienin, and ternatins bind eEF1A in a mutually exclusive manner and may have overlapping binding sites.

Conclusive evidence identifying the target of a cytotoxic compound is often provided by a resistance-conferring mutation. Recently, eEF1A was identified as the target of nannocystin A (*Krastel et al., 2015*), a hybrid polyketide/peptide natural product with no obvious structural similarity to ternatin, didemnin, or cytotrienin/ansatrienin (*Hoffmann et al., 2015*). Nannocystin-resistant HCT116 clones were obtained by chemical mutagenesis and selection, and all resistant clones contained a point mutation at Ala399 (A399V or A399T) in the *EEF1A1* gene (*Krastel et al., 2015*). We tested these cell lines for cross-resistance to our most potent ternatin variant **4**. Whereas heterozygous A399V or A399T clones were partially resistant (10-fold and 16-fold higher IC$_{50}$, respectively), cells homozygous for the A399V mutation were completely resistant to **4** at concentrations as high as 30 μM (*Figure 5a*). Consistent with these results, labeling of ectopically expressed Flag-eEF1A by photo-ternatin **5** was abrogated by the A399V mutation (*Figure 5b*). Thus, mutation of Ala399 in eEF1A likely preserves its essential cellular functions yet prevents ternatin binding.

The above results suggested that eEF1A mutations confer resistance in a recessive manner; conversely, ternatin sensitivity should be dominant. To explore this possibility further, we introduced A399V or WT eEF1A into parental or ternatin-resistant (A399V/A399V *EEF1A1*) HCT116 cells. To monitor cells across a range of eEF1A expression levels, we utilized a bicistronic lentiviral vector in which eEF1A expression was linked to a fluorescent mCherry reporter via the P2A 'self-cleaving' peptide (*Szymczak et al., 2004*). After lentiviral transduction, cells were labeled with carboxyfluorescein succinimidyl ester (CFSE) to quantify cell proliferation by FACS analysis (read as CFSE dilution). The antiproliferative effect of **4** was unaltered by expression of A399V or WT eEF1A in parental HCT116 cells. Strikingly, expression of WT (but not A399V) eEF1A in homozygous mutant cells (A399V/A399V *EEF1A1*) restored sensitivity to **4** (*Figure 5c*, *Figure 5—figure supplement 1*). Furthermore, the magnitude of resensitization correlated with the expression level of the WT eEF1A transgene, inferred from mCherry expression (*Figure 5d*, *Figure 5—figure supplement 1*). Collectively, these results indicate that eEF1A binding is necessary for the antiproliferative activity of ternatin cyclic peptides.

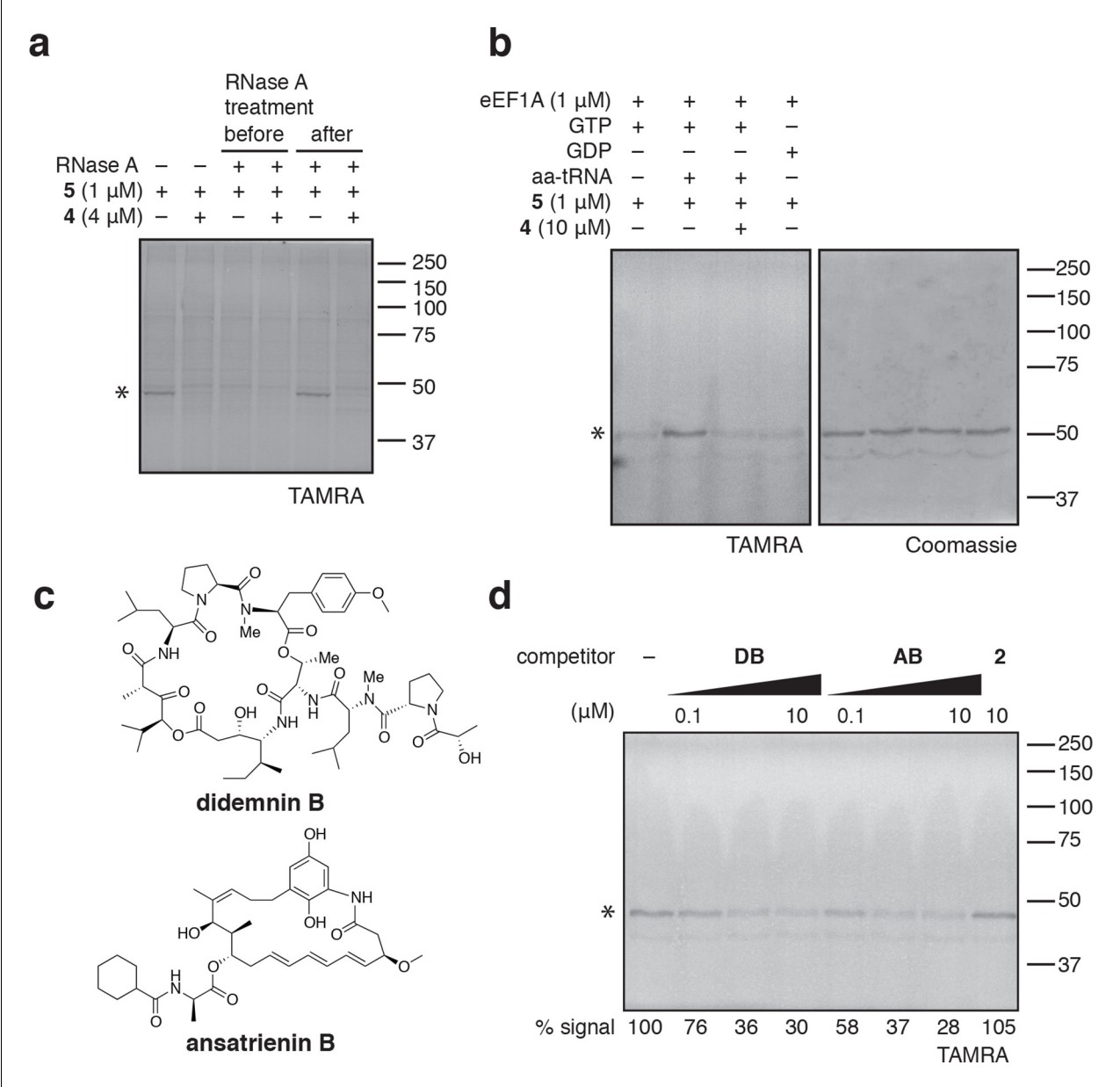

**Figure 4.** Photo-ternatin **5** binds specifically to the eEF1A ternary complex. (**a**) HEK293T cell lysates were treated with RNase A for 20 min before (lanes 3–4) or after (lanes 5–6) incubation with **5** and UV irradiation. (**b**) Purified eEF1A was incubated with GTP ± Phe-tRNA or GDP for 30 min at room temp. Reactions were treated with DMSO or **4** for 10 min, then **5** for 20 min, photolyzed and processed as in *Figure 3b*. (**c**) Translation elongation inhibitors didemnin B (DB) and ansatrienin B (AB). (**d**) A solution of eEF1A, GTP, and Phe-tRNA was incubated with the indicated compound (DB and AB: 0.1, 1.0, 10 µM) for 10 min before 20-min treatment with **5** (1 µM), followed by UV irradiation and processing as in *Figure 3b*. Coomassie-stained gels corresponding to (**a**) and (**d**) are shown in *Figure 4—figure supplement 1*.

The following figure supplements are available for figure 4:

**Figure supplement 1.** Coomassie-stained gels corresponding to (**a**) *Figure 4a*, and (**b**) *Figure 4d*.

**Figure supplement 2.** Photo-affinity labeling of HEK293T cell lysates by photo-ternatin **5** in the presence of increasing concentrations of (**a**) didemnin B (DB), or (**b**) ansatrienin B (AB).

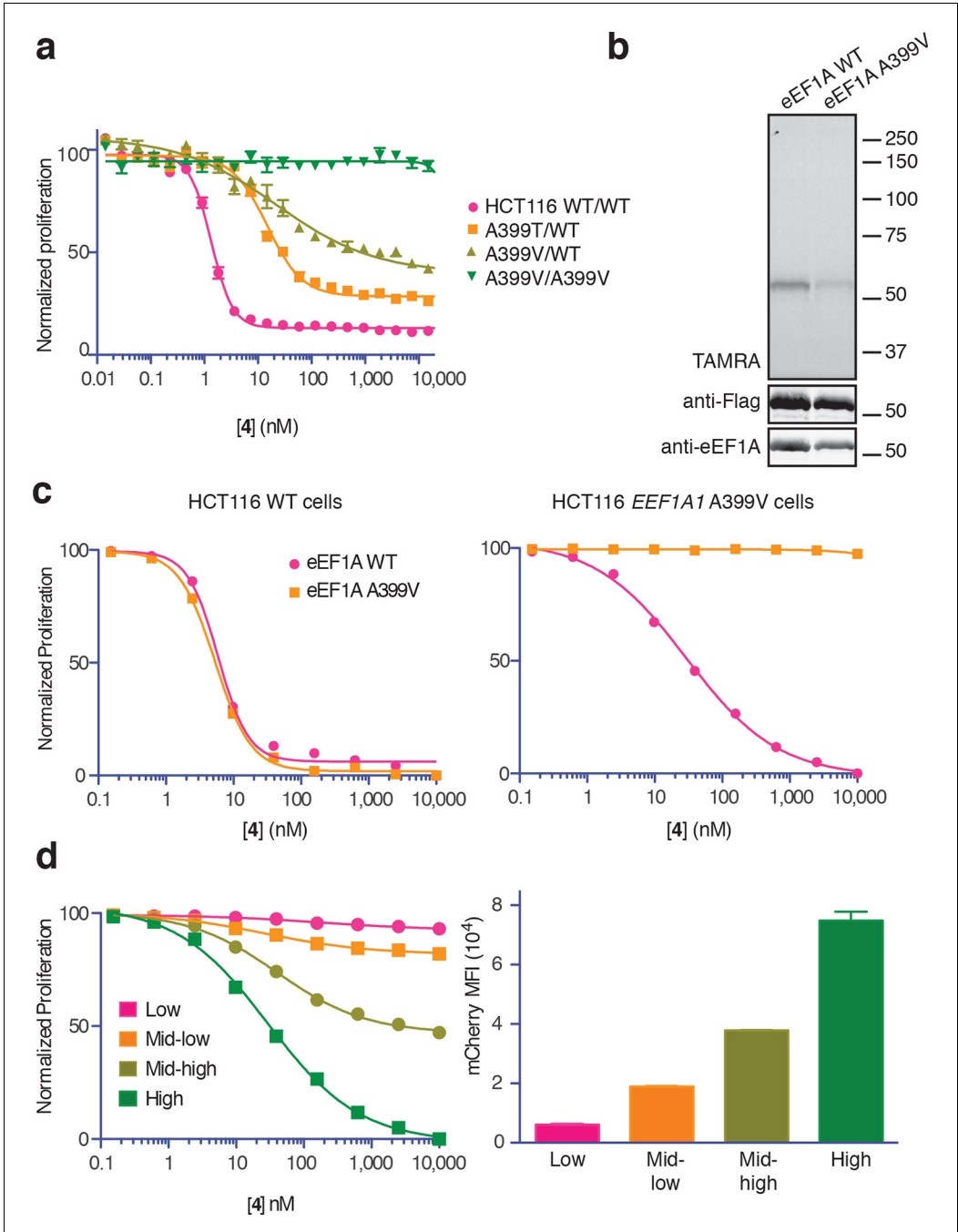

**Figure 5.** Ala399 mutation in *EEF1A1* confers resistance to **4**. (a) Effect of **4** (two-fold dilutions) on proliferation of WT and *EEF1A1*-mutant HCT116 cells over 72 hr. (b) HEK293T cells were transfected with WT or A399V Flag-eEF1A. Lysates were treated with probe **5** and processed as described in *Figure 3d*. (c) WT HCT116 cells (left) or cells homozygous for A399V *EEF1A1* (right) were transduced with a bicistronic lentiviral vector encoding eEF1A (WT or A399V) and mCherry. Cells were labeled with carboxyfluorescein succinimidyl ester (CFSE), treated with **4** for 72 hr (four-fold dilutions), and analyzed by FACS. Proliferation was assessed by CFSE dilution in high mCherry-expressing cells (mean fluorescence intensity, MFI = 6.7–7.4 × 10^4). CFSE histograms are shown in *Figure 5—figure supplement 1*. (d) Homozygous A399V *EEF1A1* HCT116 cells were transduced with WT eEF1A/mCherry as described in (c). Antiproliferative effects of **4** (left) were analyzed in cells gated according to the indicated mCherry mean fluorescence intensity (right).

The following figure supplement is available for figure 5:

**Figure supplement 1.** Ternatin sensitivity is genetically dominant.

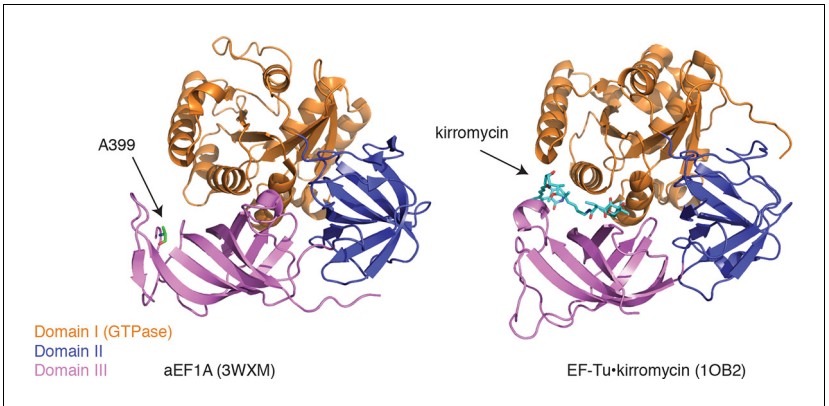

**Figure 6.** Ternatin may inhibit eEF1A by a mechanism related to kirromycin inhibition of EF-Tu. Left: crystal structure of archaeal EF1A (PDB code: 3WXM, 51% identity with human eEF1A), showing the location of A399 (human numbering) on the surface of domain III. Right: crystal structure of EF-Tu (PDB code: 1OB2, 25% sequence identity with human eEF1A), showing the kirromycin binding site at the interface of domain I and III.

Ala399, which is conserved in yeast and archaebacterial EF1A orthologs, lies on an exposed hydrophobic surface of domain III (*Figure 6*, left panel) (*Kobayashi et al., 2010*). The corresponding surface of bacterial EF-Tu forms part of the binding site for the antibiotic kirromycin (*Figure 6*, right panel) (*Vogeley et al., 2001*). Kirromycin binds to the interface of domain III and the GTPase domain I in the context of the ternary complex and stalls translation by preventing the release of EF-Tu following delivery of aa-tRNA to the ribosome A-site. Similar to the case with eEF1A and ternatin, expression of wild-type EF-Tu in a drug-resistant mutant background confers dominant sensitivity to kirromycin (*Parmeggiani and Swart, 1985*). Thus, ternatin, nannocystin, didemnin, and ansatrienin/cytotrienin may stall translation in a manner analogous to kirromycin.

## Conclusions and perspective

In this study, we sought to identify the target of the cytotoxic natural product, ternatin, previously characterized as a potent adipogenesis inhibitor. Inspired by the partially elucidated structure and cytotoxic activity of the cyclic heptapeptide natural product A3, we designed compounds **3** and **4**, which are up to 500-fold more potent than ternatin. Ternatins inhibit protein synthesis with potencies that correlate with their ability to block cell proliferation, and photo-ternatin **5** identified eEF1A as a plausible target. Surprisingly, photo-ternatin crosslinked to purified eEF1A only in the context of its ternary complex with GTP and aminoacylated tRNA. To our knowledge, specific binding of a small molecule to the eEF1A ternary complex has not been reported previously. Whether this binding event blocks translation elongation by a mechanism analogous to the antibiotic kirromycin, as suggested by the genetic dominance of ternatin sensitivity, is an exciting question that awaits future biochemical and structural studies.

Also unexpected was our discovery that two structurally unrelated natural products, didemnin and ansatrienin, compete with **5** for binding to the eEF1A ternary complex. Combined with the recent finding that nannocystin A targets domain III of eEF1A (*Krastel et al., 2015*), our study suggests that four natural products with highly diverse chemical structures have evolved independently to target a shared interaction surface near Ala399 on eEF1A. A didemnin variant, plitidepsin, is currently in advanced clinical trials for multiple myeloma and T-cell lymphoma (*PharmaMar Pipeline*). Hence, drugging this surface on eEF1A may have clinical utility. Of the four structurally divergent natural products now identified as eEF1A inhibitors, ternatins may be particularly attractive as leads for future medicinal chemistry efforts, due to their relative structural simplicity and accessibility by total synthesis.

# Materials and methods

## Cell culture

HCT116 cells (ATCC, Manassas, VA) were maintained in McCoy's 5A media (Gibco, Grand Island, NY) supplemented with 10% fetal bovine serum (Axenia Biologix, Dixon, CA), 100 units/mL penicillin, and 100 ug/mL streptomycin (Gibco). Mutant HCT116 cells were obtained from Dominic Hoepfner (*Krastel et al., 2015*). HEK293T cells (ATCC) and HeLa cells (ATCC) were maintained in DMEM (Gibco) supplemented with 10% fetal bovine serum, 100 units/mL penicillin, and 100 μg/mL streptomycin. All other cell lines were obtained from the Cancer Cell Line Encyclopedia collection and were maintained as described (*Barretina et al., 2012*). All cell lines were tested for mycoplasma (Lonza, CH) before use and were SNP-typed to verify that their identity matched that of the originally procured sample. None of the tested lines are present in the list of cross-contaminated/misidentified cell lines (http://iclac.org/wp-content/uploads/Cross-Contaminations-v7_2.pdf). All cells were cultured at 37°C in a 5% $CO_2$ atmosphere.

## Proliferation assay (*Figure 1b*, *3b*, *Figure 3—figure supplement 1*)

Growing cells were briefly trypsinized and repeatedly pipetted to produce a homogenous cell suspension. 2,500 cells were seeded in 100 μL complete growth media per well in 96-well clear-bottom plates. After allowing cells to adhere overnight, cells were treated with 25 μL/well 5X drug stocks (0.1% DMSO final) and incubated for 72 hr. AlamarBlue (Life Technologies, Grand Island, NY) was used to asses cell viability per the manufacturer's instructions. Briefly, 12.5 μL alamarBlue reagent was added to each well, and plates were incubated at 37°C. Fluorescence intensity was measured hourly to determine the linear range for each assay (Ex 545 nm, Em 590 nm, Spectra Max M5, Molecular Devices, Sunnyvale, CA). Proliferation curves were generated by first normalizing fluorescence intensity in each well to the DMSO-treated plate average. Normalized fluorescence intensity was plotted in GraphPad Prism (GraphPad, La Jolla, CA), and $IC_{50}$ values calculated from nonlinear regression curves. The reported $IC_{50}$ values represent the average of at least three independent determinations (±SEM).

## Proliferation assay (*Figure 1c*, *5a*)

Proliferation assays with the 21-cell line panel (*Figure 1c*) and *EEF1A1* mutant and parental HCT116 cells (*Krastel et al., 2015*) were performed using an ultra-high throughput screening system (GNF Systems, San Diego, CA). For the 21-cell line panel, cells were harvested and suspended at a concentration of 50,000 cells/mL in the appropriate medium. Cells (250 cells/well, 5 μL/well) were then dispensed into Greiner (Monroe, NC) white, solid-bottom, TC-treated, 1536-well assay plates (Griener # 789173-A) and incubated for 10 hr at 37°C (95% humidity, 5% $CO_2$). Compounds (15 nL/well, 16-point two-fold dilution series) and controls (15 nL/well, MG-132 and DMSO) were then added to the assay plates (n = 4) using an Echo acoustic liquid dispenser (Labcyte, Sunnyvale, CA). Assay plates were incubated 3 days at 37°C (95% humidity, 5% $CO_2$) before addition of CellTiter Glo (4 μL/well, Promega, Madison, WI). Assay plates were incubated for 15 min at room temperature and then luminescence was measured using a ViewLux uHTS Microplate Imager (PerkinElmer, Waltham, MA). Reported $IC_{50}$ values were calculated as described (*Barretina et al., 2012*). Experiments with *EEF1A1* mutant and parental HCT116 cells were conducted in the same manner as described above, except compounds were applied by 22-point two-fold dilution series (30 μM–0.1 nM).

## eEF1A-P2A-mCherry cell line generation

WT and A399V eEF1A coding sequences were inserted by Gibson assembly (*Gibson et al. 2009*) into a pHR lentiviral vector containing an eEF1A promoter and P2A-mCherry sequence (gift of A. Weiss, UCSF). Lentiviral particles were generated as described (http://www.broadinstitute.org/rnai/public/resources/protocols). Briefly, HEK293T packaging cells in 6-well plates were transfected with eEF1A-P2A-mCherry expression plasmid, 2nd-generation packaging plasmid pCMV-dR8.91, and envelope plasmid pMD2.G using TransIT-LT1 transfection reagent (Mirus Bio, Madison, WI) according to the manufacturer's instructions. After 24 hr, media was replaced with viral harvest media (complete growth media +12 mg/mL BSA). Virus-containing media was collected after 24 and 48 hr, pooled, and used for infections without further processing. Parental and *EEF1A1* A399V

homozygous mutant HCT116 cells at 70% confluence in 6-cm dishes were treated with polybrene (8 µg/mL, AmericanBio, Natick, MA) and virus (0.5 mL). After 24 hr, cells were trypsinized and passaged in complete growth media.

## Proliferation assay, CFSE dilution (*Figure 5c,d*)

Trypsinized cells ($8.6 \times 10^5$) were washed with PBS, resuspended in 1 mL PBS, and treated with CFSE (1 mL, 5 µM in PBS, Life Technologies). After 5 min, FBS (2 mL) was added, and cells were washed once with complete growth media. CFSE-labeled cells were plated ($8 \times 10^4$ cells/well) in 6-well dishes and treated with **4** or DMSO. After 72 hr, plates were centrifuged (500 ×g, 5 min), media aspirated, and cells trypsinzied. Trypsinized cells were washed with FACS buffer (PBS –Mg/Ca + 2% FBS + 2 mM EDTA) and resuspended in the same buffer (0.4 mL). Samples were analyzed by flow cytometry (BD LSRFortessa, BD, Franklin Lakes, NJ), with data analysis using FlowJo software (Tree Star, Ashland, OR). mCherry expression gates were defined to span a ten-fold expression range with at least 500 events for each data point.

## $^{35}$S-methionine incorporation

HCT116 cells at 80% confluence in 12-well plates were incubated with compounds for 5 hr at 37°C. The cell culture media was replaced with methionine- and cysteine-free DMEM with the appropriate amount of compound. $^{35}$S-methionine (PerkinElmer EasyTag, 40 µCi/well) was added and cells were incubated for 1 hr. Subsequently, media was aspirated and cells were washed twice with ice-cold phosphate-buffered saline (PBS). For samples analyzed by SDS-PAGE/autoradiography (*Figure 2a*), cells were lysed in buffer (PBS, 0.5% Triton X-100, Roche complete protease inhibitor cocktail, Roche, Indianapolis, IN) and protein content was normalized by Bradford assay. Dried gels were exposed to a storage phosphor screen (Amersham Biosciences, UK) overnight and scanned with a Typhoon imager (GE Life Sciences, Pittsburgh, PA). Bulk incorporation measurements (*Figure 2b*) were made by treating cells with cold 10% trichloroacetic acid (TCA, 0.5 mL) for 10 min. Precipitated monolayers were washed with 10% TCA ($2 \times 0.5$ mL) and solubilized in 0.2 N NaOH (0.5 mL). Aliquots (100 µL) were diluted into liquid scintillation fluid and radioactivity was quantified by scintillation counting.

## Ribosome/polysome profiles

HeLa cells grown to 80% confluence in 10-cm dishes were treated with DMSO or **4** (5 µM) for 20 min (*Figure 2c*, left panel), or were first treated with DMSO, **4** (5 µM), or cycloheximide (CHX, 100 µg/mL, Sigma, St. Louis, MO) for 15 min, followed by harringtonine (2 µg/mL, LKT Laboratories, St. Paul, MN) for 20 min. Following treatment, dishes were placed on ice, media aspirated, and cells were washed once with ice-cold PBS. Cells were scraped into 400 µL lysis buffer (20 mM Tris-HCl pH 7.5, 150 mM NaCl, 5 mM MgCl$_2$, 1 mM DTT, 1% Triton X-100, 100 µg/mL CHX). Lysates were incubated for 10 min on ice, then passed through a 26-G needle 5 times. Samples were clarified by centrifugation at 14,000 rpm for 10 min at 4°C. Lysates were applied to 10–50% sucrose gradients (prepared in 20 mM Tris-HCl pH 7.5, 150 mM NaCl, 5 mM MgCl$_2$, 1 mM DTT, 100 µg/mL CHX), followed by centrifugation at 35,000 rpm for 3 hr in a SW41 rotor at 4°C. Ribosome/polysome profiles were obtained by flowing the gradient top to bottom through an in-line spectrometer (254 nm, EM-1 Econo UV Monitor, BioRad, Hercules, CA).

## Lysate preparation for photo-labeling

Lysates were prepared from HEK293T cells by washing a monolayer of cells once with ice-cold PBS and briefly freezing at –80°C. Lysis buffer (25 mM HEPES-KOH pH 7.3, 100 mM KOAc, 5 mM MgOAc$_2$, 1 mM TCEP, 1 mM PMSF, Roche EDTA-free complete protease inhibitor) was added, and cells were detached with a cell scraper. Crude lysates were clarified by centrifugation (15 min at 14,000 rpm), protein was quantified by Bradford assay, and the protein concentration was brought to 1.5 mg/mL with lysis buffer. Lysates were processed for photo-labeling as described below. For samples treated with RNase A, RNase A (Roche) was added to lysates (10 mg/mL final concentration) and incubated for 20 min at RT.

### Photo-labeling, general procedure

Photo-labeling experiments were preformed essentially as described (*MacKinnon and Taunton, 2009*; *MacKinnon et al., 2007*). Reactions (100 µL) were treated with DMSO (1 µL) or competitor in DMSO as 100X stocks and incubated for 10 min at RT, then treated with photo-probe **5** and incubated for 20 min. Reactions were transferred to 96-well plates and irradiated with long-wave UV light ($\lambda_{max}$ = 355 nm, 1000 W, 90 s). To each reaction (34 µL) was added a click chemistry master mix (8.3 µL) made from the following stock solutions: 1.7 mM TBTA in 1:4 DMSO/$^t$BuOH [2.5 µL]; 5 mM TAMRA-N$_3$ [0.5 µL]; 50 mM TCEP [0.8 µL, freshly prepared]; 10% SDS [4.5 µL]. Then, CuSO$_4$ (0.8 µL, 50 mM stock), was added and reactions were incubated for 1 hr at RT. Reactions were quenched with 5X Laemmli sample buffer, and separated by SDS-PAGE. Gels were scanned for TAMRA fluorescence (532 nm excitation laser, 568 nm emission measured with LPG filter) on a Typhoon FLA 9000 scanner (GE Life Sciences).

### Flag immunoprecipitation and western blotting

HEK293T cells were transfected overnight with Flag-eEF1A DNA (pQCXIP-Flag-eEF1A1, provided by Orna Elroy-Stein [*Sivan et al., 2011*]) in 6-well dishes using Lipofectamine 2000 (Invitrogen, Grand Island, NY) according to the manufacturer's instructions. Flag-eEF1A1 A399V was produced by PCR amplification of the plasmid with the following primers:

> 5'-CTTGAAGTCTGGTGATGTTGCCATTGTTGATATGGTTCC-3'
> 5'-GGAACCATATCAACAATGGCAACATCACCAGACTTCAAG-3'

Lysates were prepared, treated with **5** ± competitor, and photolyzed as described above. After photolysis, samples were immunoprecipitated overnight at 4°C with anti-Flag magnetic beads (Sigma). Beads were washed four times with buffer (50 mM Hepes pH 7.5, 150 mM NaCl, 0.5% Triton X-100), then eluted (5 min, 100°C) in 1% SDS, 25 mM Hepes pH 7.5. Eluates were subjected to click chemistry with TAMRA-N$_3$ and processed for in-gel fluorescence scanning as described above.

Gels were transferred to nitrocellulose by the semi-dry method, blocked with Odyssey Blocking Buffer (LI-COR Biosciences, Lincoln, NE), incubated with anti-EF1A (Millipore, DE, cat. 05–235) or anti-Flag (Sigma, cat. F3165), and detected with goat anti-mouse light chain-specific secondary antibody conjugated to 680 nm near-IR dye (Jackson ImmunoResearch Laboratories, West Grove, PA). Blots were scanned on an Odyssey infrared imager (LI-COR Biosciences).

### Reconstitution of eEF1A ternary complex

Native eEF1A was purified from rabbit reticulocytes as described (*Carvalho et al., 1984*). Bulk rabbit reticulocyte tRNA was purified as described (*Merrick, 1979*). tRNA was charged with phenylalanine using a crude reticulocyte tRNA synthetase preparation (*Merrick, 1979*) as described (*Walker and Fredrick, 2008*). Reactions (100 µL) were assembled by adding the following components in order: 50 µL 2X buffer (100 mM HEPES pH 7.3, 250 mM KOAc, 10 mM MgCl$_2$), TCEP (1 µL,100 mM), GTP or GDP (1 µL,10 mM), ± Phe-tRNA (2 µL, ~0.1 mM), water to 97 µL, and eEF1A to 1 µM. Samples were incubated for 20 min at RT before processing for photo-labeling as described above. Ansatrienin B was purchased from Enzo Life Sciences (Farmingdale, NY) and prepared as a 10 mM stock in DMSO. Didemnin B was obtained from the NCI/DTP chemical repository and was prepared as a 10 mM stock in DMSO.

## Chemical synthesis

### General

Materials purchased from commercial vendors were of reagent grade and used without further purification. (*R*)-2,5-Dihydro-3,6-dimethoxy-2-isopropylpyrazine (Schöllkopf auxiliary) was purchased from Green Chempharm Inc (Bardonia, NY). *N*-Boc amino acids were purchased from Chem-Impex International (Wood Dale, IL). All other reagents were purchased from Sigma-Aldrich unless otherwise noted. (*S*)-Boc-photo-leucine was prepared from Boc-4,5-dehydro-Leu-OH (Chem-Impex Intl.) as described (*MacKinnon et al., 2007*). (2*R*,3*R*)-ethyl 2-amino-3-hydroxy-4-methylpentanoate (*Shimokawa et al., 2007*) (**11**, *Figure 8*), tripeptide **17** (*Figure 8*) (*Shimokawa et al., 2009*), and iodides **6a** and **6b** (*Nakamura and Mori, 2000*; *Smith et al., 1996*) were prepared as described.

**Figure 7.** Synthesis of (2S, 4R)- and (2S, 4S)-methyl 2-amino-4-methylhex-5-enoate (dehydro-homoleucine). Reagents and conditions: (a) $^n$BuLi, THF, –78°C; (b) TBAF, THF, 0°C; (c) Dess-Martin periodinane, $CH_2Cl_2$; (d) MePPh$_3$Br, $^n$BuLi, THF, 0°C; (e) TFA, $H_2O$/MeCN.

Water and air sensitive reactions were conducted in flame-dried glassware under an inert argon atmosphere. Dry solvents were prepared on a Glass Contour (Laguna Beach, CA) system, which dispenses solvents through either alumina or activated molecular sieve columns. $^1$H and $^{13}$C NMR spectra were obtained on a Varian (Palo Alto, CA) Inova 400 MHz spectrometer, a Bruker (Billerica, MA) Avance DRX500 spectrometer equipped with a QCI-Cryoprobe, or a Bruker AvanceIII 600 spectrometer, and referenced to the residual solvent peak. Spectral data are reported as follows: chemical shift (in ppm), integration, multiplicity (br, broad; s, singlet; d, doublet; t, triplet; q, quartet; m, multiplet; or as a combination of these), and coupling constant(s) in Hertz. LC-MS analysis was performed on a Waters (Milford, MA)

Acquity LCT UPLC equipped with a TUV detector (monitored at 215 nm) and a Waters Acquity UPLC 1.7 μm C-18 column, eluting with a water/MeCN (+0.1% formic acid) gradient at 0.6 mL/min. Purification by reverse-phase HPLC was performed on either a Varian ProStar 210 purification system with a UV-Vis detector (model 345) monitoring at 215 nm, or a Waters binary gradient purification system (model 2545) with a photodiode array detector (model 2998).

Two dehydro-homolecuine diastereomers were prepared according to the scheme in *Figure 7*.

### (2S,5R)-2-((R)-3-((*tert*-butyldimethylsilyl)oxy)-2-methylpropyl)-5-isopropyl-3,6-dimethoxy-2,5-dihydropyrazine (7a)

To a solution of Schöllkopf auxiliary (7 g, 38 mmol) in dry THF (230 mL) at –78°C was added $^n$BuLi (2.5 M in THF, 14.4 mL, 36 mmol) dropwise. The reaction was stirred for 1 hr at –78°C, then **6a** (6 g, 20 mmol in 40 mL THF) was added dropwise. The reaction was kept overnight at –78°C, then quenched with saturated aqueous NH$_4$Cl. The organic and aqueous phases were separated, and the aqueous layer was extracted with EtOAc (3 × 40 mL). The combined organic layers were washed with brine, dried (MgSO$_4$), filtered, and concentrated. The resulting clear yellow oil was purified by flash column chromatography (1–10% EtOAc in hexanes) to provide **7a** as a clear oil (4.4 g, 59%). $^1$H NMR (400 MHz, CDCl$_3$): δ 0.03 (6H, s), 0.69 (3H, d, J = 6.8 Hz), 0.89 (9H, s), 0.90 (3H, d, J = 10 Hz), 1.05 (3H, d, J = 6.8 Hz), 1.39 (1H, m), 1.89 (2H, m), 2.27 (1H, 7d, J = 3.4, 6.8 Hz), 3.40 (1H, dd, J = 6.0, 10.0 Hz), 3.52 (1H, dd, 4.8, 10.0 Hz), 3.67 (3H, s), 3.68 (3H, s), 3.93 (1H, m), 4.04 (1H, m). $^{13}$C NMR (100 MHz, CDCl$_3$): δ 5.2, 16.7, 17.6, 18.5, 19.2, 26.1 (3C), 31.8, 32.3, 38.2, 52.4, 52.5, 54.3, 60.6, 68.2, 94.6, 163.3, 164.8. LC-MS (ESI +): 371.44 (+).

### (2S,5R)-2-((S)-3-((*tert*-butyldimethylsilyl)oxy)-2-methylpropyl)-5-isopropyl-3,6-dimethoxy-2,5-dihydropyrazine (7b)

In the same manner as above, **6b** (2.8 g, 8.8 mmol) was converted to **7b** (2.1 g, 64%). $^1$H NMR (400 MHz, CDCl$_3$): δ 0.03 (6H, s), 0.69 (3H, d, J = 6.8 Hz), 0.88 (9H, br), 0.94 (3H, d, J = 6.8 Hz), 1.05 (3H,

d, $J$ = 6.8 Hz), 1.58 (2H, m), 1.96 (1H, br), 2.27 (1H, 7d, $J$ = 3.2, 6.8 Hz), 3.35 (1H, dd, $J$ = 6.8, 10 Hz), 3.47 (1H, dd, $J$ = 6, 10 Hz), 3.67 (3H, s), 3.68 (3H, s), 3.91 (1H, m), 4.02 (1H, m). $^{13}$C NMR (100 MHz, CDCl$_3$): δ 5.2, 16.7, 16.8, 18.5, 19.2, 26.1 (3C), 31.7, 32.4, 38.4, 52.4, 52.5, 53.9, 60.7, 68.9, 94.6, 163.1, 164.8. LC-MS (ESI +): 371.44 (+).

## (R)-3-((2S,5R)-5-isopropyl-3,6-dimethoxy-2,5-dihydropyrazin-2-yl)-2-methylpropan-1-ol (8a)

To a solution of **7a** (4.8 g, 12.9 mmol) in THF (130 mL) cooled to 0°C, TBAF (1 M in THF, 15.5 mL, 15.5 mmol) was added slowly. The reaction was warmed to room temperature, stirred overnight, and then saturated aqueous NH$_4$Cl (15 mL) was added. The aqueous layer was extracted with EtOAc (4 × 20 mL). The combined organic layers were washed with brine (20 mL), dried (MgSO$_4$), filtered, and concentrated. The resulting oil was purified by flash column chromatography on Et$_3$N-treated

**Figure 8.** Synthesis of ternatin variants. Reagents and conditions: (a) HATU, DIPEA, DCM/DMF; (b) 2 M HCl, MeOH, 30°C; (c) 1 N LiOH, H2O/THF; (d) HATU, DIPEA, DMF.

silica (7:3 hexanes/EtOAc + 3% Et₃N) to provide **8a** as a clear oil(1.3 g, 87%). [1]H NMR (400 MHz, CDCl₃): δ 0.71 (3H, d, J = 6.8 Hz). 0.99 (3H, d, J = 6.8 Hz), 1.04 (3H, d, J = 6.8 Hz), 1.63 (1H, m), 1.98 (1H, m), 2.07 (1H, m), 2.25 (1H, m) 3.49 (1H, m), 3.57 (1H, m), 3.69 (3H, s), 3.70 (3H, s), 3.99 (1H, t, J = 3.6 Hz), 4.05 (1H, m), 4.29 (1H, m). [13]C NMR (100 MHz, CDCl₃): δ 16.8, 16.9, 19.2, 32.1, 32.9, 39.2, 52.5, 52.7, 53.0, 61.1, 67.5, 164.1, 164.7. LC-MS (ESI +): 257.36 (+).

## (S)-3-((2S,5R)-5-isopropyl-3,6-dimethoxy-2,5-dihydropyrazin-2-yl)-2-methylpropan-1-ol (8b)

In the same manner as above, **7b** (1.8 g, 4.8 mmol) was converted to **8b** (1.1 g, 89%). [1]H NMR (400 MHz, CDCl₃): δ 0.69 (3H, d, J = 6.8 Hz), 0.93 (3H, d, J = 6.8 Hz), 1.05 (3H, d, J = 6.8 Hz), 1.59 (1H, br), 1.94 (1H, m), 2.18–2.30 (2H, m), 3.36 (1H, m), 3.57 (1H, m), 3.69 (3H, s), 3.70 (3H, s), 3.94–3.99 (2H, m), 5.66 (1H,br). [13]C NMR (100 MHz, CDCl₃): δ 16.8, 19.1, 19.2, 32.0, 37.1, 42.3, 52.8, 53.2, 55.8, 60.9, 68.9, 163.5, 164.6. LC-MS (ESI +): 257.35 (+).

## (2R,5S)-2-isopropyl-3,6-dimethoxy-5-((R)-2-methylbut-3-en-1-yl)-2,5-dihydropyrazine (9a)

To a solution of Dess-Martin periodinane (5.9 g, 14 mmol) in wet CH₂Cl₂ (55 mL), **8a** (2.9 g, 11.1 mmol) in CH₂Cl₂ (110 mL) was added slowly. After 1 hr, NaOH(1.5 M, 70 mL) was added and the solution was stirred for 10 min. Layers were separated and the organic layer was shaken with NaOH (0.75 M, 70 mL). The combined aqueous layers were extracted with Et₂O (3 × 25 mL). The combined organics were dried (MgSO₄), filtered, and concentrated. The resulting aldehyde was azeotropically dried (3 × MeCN) then carried on to the next step without purification.

$^n$BuLi (2.5 M in hexanes, 9.3 mL, 23 mmol) was added to oven-dried methyl triphenylphosphonium bromide (8.8 g, 24.5 mmol) in dry THF (170 mL) at 0°C, and stirred for 30 min until a deep yellow color persisted. The crude aldehyde (2.8 g, 11 mmol) in dry THF (25 mL) was added dropwise at 0°C, then stirred for 30 min. The reaction was quenched with saturated aqueous NH₄Cl. The aqueous layer was extracted with EtOAc (3 × 20 mL). Combined organics were washed with brine, dried (MgSO₄), filtered, and concentrated. The resulting oil was purified by flash column chromatography (1–5% EtOAc in hexanes) to give **9a** as a clear oil (1.1 g, 40%). [1]H NMR (400 MHz, CDCl₃): δ 0.68 (3H, d, J = 6.8 Hz), 0.99 (3H, d, J = 7.2 Hz), 1.04 (3H, d, J = 7.2 Hz), 1.48 (1H, m), 1.85 (1H, m), 2.26 (1H, m), 2.47–2.56 (1H, br), 3.67 (3H, s), 3.69 (3H, br), 3.89 (1H, t, J = 3.4 Hz), 3.97 (1H, dt, J = 3.6, 8.8 Hz) 4.90 (2H, m), 5.71 (1H, m). [13]C NMR (100 MHz, CDCl₃): δ 16.8, 19.2, 21.3, 31.8, 34.4, 41.6, 52.4, 52.5, 54.0, 60.9, 113.1, 114.3, 162.9, 164.6. LC-MS (ESI +): 253.34 (+).

## (2R,5S)-2-isopropyl-3,6-dimethoxy-5-((S)-2-methylbut-3-en-1-yl)-2,5-dihydropyrazine (9b)

In the same manner as above, **8b** (1.1 g, 4.3 mmol) was converted to **9b** (450 mg, 42%).[1]H NMR (400 MHz, CDCl₃): δ 0.69 (3H, d, J = 6.8 Hz), 1.03 (3H, d, J = 6.8 Hz), 1.04 (3H, d, J = 6.8 Hz), 1.64 (1H, m), 1.78 (1H, m), 2.27 (1H, m), 2.49 (1H, m), 3.68 (3H, s), 3.69 (3H, s), 3.91 (1H, t, J = 3.2 Hz), 4.01 (1H, m), 4.86 (1H, m), 4.93 (1H, m), 5.71 (1H, m). LC-MS (ESI +): 253.34 (+).

## (2S,4R)-methyl 2-amino-4-methylhex-5-enoate (10a)

To a solution of **9a** (550 mg, 2.1 mmol) in MeCN (10 mL), TFA$_{aq}$ (0.1 N, 15 mL) was added and the reaction stirred overnight. The reaction mixture was evaporated and the resulting white solid was purified by reverse-phase HPLC. Column: Waters XBridge Prep C18, 5 μm, 30 × 250 mm. Conditions: 0–25% MeCN/H₂O (+ 0.1% formic acid in both mobile phases) linear gradient over 40 min, flow rate 20 mL/min. The purified formate salt was dissolved in saturated aqueous NaHCO₃ (5 mL), and then the aqueous solution was extracted with Et₂O (4 × 10 mL). The organic fractions were combined and HCl (0.6 mL, 4 N in dioxane) was added. Solvent was evaporated to give **10a** as a white solid (140 mg, 0.7 mmol, 34%). [1]H NMR (400 MHz, DMSO-d6): δ 0.98 (3H, d, J = 6.8 Hz), 1.66 (1H, ddd, J = 6.0, 8.0, 14.0 Hz), 1.77 (1H, ddd, J = 5.6, 8.8, 14.4), 2.35 (1H, m), 3.73 (3H, s), 3.82 (1H, dd, J = 5.6, 8.0 Hz), 5.02 (1H, m), 5.09 (1H, m), 5.67 (1H, m), 8.14 (3H, br). [13]C NMR (100 MHz, DMSO-d6): δ 19.8, 33.2, 37.2, 50.7, 52.7, 114.7, 142.3, 170.7. LC-MS (ESI +): 158.26 (+).

### (2S,4S)-methyl 2-amino-4-methylhex-5-enoate (10b)

In the same manner as above, **9b** (300 mg, 1.2 mmol) was converted to **10b** (168 mg, 75%). $^1$H NMR (400 MHz, DMSO-$d6$): δ 0.98 (3H, d, $J$ = 6.8 Hz), 1.74 (2H, t, $J$ = 6.8 Hz), 2.33 (1H, p, $J$ = 7.2 Hz), 3.75 (3H, s), 3.94 (1H, t, $J$ = 6.8 Hz), 5.03 (2H, m), 5.66 (1H, m), 8.28–8.40 (3H, br). $^{13}$C NMR (100 MHz, DMSO-$d6$): δ 19.9, 33.1, 37.0, 50.5, 52.6, 114.4, 142.4, 160.3. LC-MS (ESI +): 159.23 (+).

## Synthesis of cyclic peptides

Ternatin (**1**) and ternatin-4-Ala (**2**) were synthesized as described, and all spectral data were in accordance with those reported (*Shimokawa et al., 2008b*). Cyclic peptides **3**, **4**, **4b**, and **5**, were synthesized by the same route as ternatin (*Figure 8*).

### General procedure for coupling Boc-protected amino acids (a)

A flask was charged with the *N*-Boc amino acid (1 equiv), *N*-deprotected peptide methyl ester (1 equiv), HATU (1.1 equiv), and CH$_2$Cl$_2$/DMF (1:1 v/v, 0.1 M peptide methyl ester). The reaction was cooled to 0°C, and *N,N*-diisopropylethylamine (3 equiv) was added dropwise, giving a clear yellow solution. The reaction was allowed to warm to room temperature, and was monitored by LC-MS. After 3–12 hr, the reaction was quenched by the addition of saturated aqueous NH$_4$Cl (1–5 mL) and diluted with EtOAc (10–20 mL). The organic layer was separated, and the aqueous layer was extracted with EtOAc (4 × 10–20 mL). The organic fractions were combined and washed with brine (10–20 mL), dried (MgSO$_4$), filtered, and concentrated. Peptides were purified by silica gel column chromatography using hexane/EtOAc gradients (dipeptides, tripeptides, tetrapeptides), or isopropyl alcohol/toluene gradients (linear heptapeptides).

### General procedure for Boc deprotection (b)

The *N*-Boc amino ester (1 equiv, 0.5 M in MeOH) was treated with HCl (5 equiv, 2 M in MeOH) and gently warmed (30–35°C). Reaction progress was monitored by LC-MS until no starting material remained (2 hr). Solvent was evaporated, and the residue was azeotropically dried with MeCN until the product appeared as a white solid, which was used for the next coupling step without purification.

### General procedure for ester hydrolysis (c)

The ester was dissolved in 3:1 THF/H$_2$O (0.1 M) and was treated with 1 N LiOH (10 equiv). Reaction progress was monitored by LC-MS until no starting material remained (1–2 hr). The reaction was brought to neutral pH with stoichiometric 1 N HCl and solvent was evaporated. The resulting residue was azeotropically dried with MeCN and used in the next coupling step without further purification.

### General procedure for macrocyclization (d)

Macrocyclization was carried out under a pseudo-high dilution protocol using two syringe pumps carrying (1) the fully deprotected linear heptapeptide in DMF (1 equiv, 0.02 M), and (2) HATU in DMF (3 equiv, 0.1 M), which were added at 0.01 mL/min to a flask containing DIPEA in DMF (6 equiv, 0.1 M). After addition, the reaction was allowed to stir at room temperature for 24 hr, then the solvent was distilled under reduced pressure (bath temperature <30°C). The resulting residue was dissolved in EtOAc (10 mL), washed with 1 N HCl (5 mL), saturated aqueous NaHCO$_3$ (5 mL), and brine (2 mL), then dried over MgSO$_4$, filtered, and concentrated. The resulting residue was dissolved in DMSO and purified by reverse phase HPLC. Column: Peeke Scientific Combi-A 5 μm preparative C18, 50 × 22 mm. Conditions: MeCN/H$_2$O 40–45% (+ 0.1% TFA in both mobile phases) linear gradient over 20 min, flow rate 10 mL/min.

**Figure 9.** Compound 12.

[1]H NMR (600 MHz, DMSO-d6): δ 0.77 (3H, d, J = 7.2 Hz), 0.89 (3H, J = 6.6 Hz), 1.09 (1H, d, J = 6.53 Hz), 1.15–1.19 (5H, m), 1.37 (2H, m), 1.41 (9H, s), 1.52–1.63 (3H, m), 1.76 (1H, m), 2.61 (1H, m), 2.68 (3H, s), 3.41–3.67 (2H, m), 4.01–4.11 (2H, m), 4.28–4.38 (1H, m), 4.83–5.05 (3H, m), 7.91 (1H, d, J = 8.2 Hz). [13]C-NMR (150 MHz, DMSO-d6): δ 14.5, 15.0, 16.7, 20.1, 28.5, 29.7, 38.7, 42.6, 50.5, 51.9, 55.7, 60.2, 60.6, 75.2, 75.4, 79.7, 80.0, 154.9, 170.7, 171.3, 171.4. LC-MS (ESI –): 470.36 (–).

**Figure 10.** Compound 13.

[1]H NMR (600 MHz, CDCl₃): major conformer δ 0.83–0.97 (24H, m), 1.19 (1H, td, J = 7.2, 14.2), 1.27–1.82 (20H, m), 1.46 (9H, s), 1.75 (1H, td, J = 7.2, 13.4), 2.78 (3H, s), 2.79 (3H, s), 2.95 (3H, s), 3.23–3.39 (1H, m), 3.71 (3H, s), 3.90 (1H, m), 4.06 (1H, d, J = 7.2), 4.41 (1H, q, J = 6.1), 4.48–4.56 (1H, m), 4.74 (1H, t, J = 6.8), 4.99 (1H, q, J = 6.3), 5.05 (1H, t, J = 7.8), 5.20 (1H, s), 5.49 (1H, q, J = 6.4), 6.86 (1H, d, J = 8.4), 6.91 (1H, d, J = 7.8), 7.55 (1H, d, J = 7.8). [13]C-NMR (150 MHz, DMSO-d6): major conformer δ 11.8, 14.2, 14.6, 17.1, 19.6, 20.0, 21.7, 21.8, 22.8, 23.2, 24.7, 25.0, 25.6, 26.5, 28.4, 29.6, 29.7, 30.2, 30.3, 31.0, 36.0, 36.7, 38.6, 41.1, 43.1, 50.3, 51.2, 52.3, 52.6, 52.7, 54.2, 54.9, 58.6, 60.4, 68.2, 76.9, 80.3, 155.8, 170.1, 171.0, 171.2, 171.7, 172.0, 172.1, 173.0. LC-MS (ESI –): 894.65 (–).

**Figure 11.** Compound 14.

[1]H NMR (600 MHz, CDCl₃): major conformer δ 0.99 (3H, d, J = 6.7 Hz), 1.26 (3H, d, J = 6.6 Hz), 1.44 (9H, s), 0.82–1.80 (33H, m), 1.94 (1H, m), 2.06 (1H, m), 2.17 (1H, m), 2.26 (1H, m), 2.78 (3H, s), 2.79 (3H, s), 2.94 (3H, s), 3.11 (1H, s), 3.70 (3H, s), 3.88 (1H, m), 4.37–4.47 (3H, m), 4.73 (1H, dd, J = 5.5, 8.3 Hz), 4.89–4.99 (3H, m), 5.03 (1H, dd, J = 6.7, 8.9 Hz), 5.19 (1H, br), 5.50 (1H, m), 5.59 (1H, m), 6.45 (1H, d, J = 7.9 Hz), 6.87 (1H, d, J = 8.4 Hz), 7.63 (1H, d, J = 7.6 Hz). [13]C NMR (150 MHz,

CDCl$_3$): major conformer δ 11.90, 14.2, 14.3, 14.7, 19.7, 20.8, 21.9, 22.7, 23.3, 24.8, 25.7, 26.6, 26.7, 28.4 (3C), 29.7, 29.8, 30.2, 30.5, 31.7, 35.1, 36.2, 36.8, 38.7, 38.9, 43.2, 50.4, 50.9, 51.5, 52.4, 52.6, 52.8, 53.5, 55.0, 79.2, 114.8, 142.3, 152.0, 170.2, 171.1, 171.2, 171.8, 172.1, 172.1, 172.8. LC-MS (ESI +): 908.73 (+).

**Figure 12.** Compound 15.

   [1]H NMR (600 MHz, CDCl$_3$): major conformer δ 0.86 (3H, d, $J$ = 6.5 Hz), 0.97 (3H, d, $J$ =6.7 Hz), 1.25 (3H, d, $J$ = 6.8 Hz), 1.44 (9H, s), 0.86–2.11 (32H, m), 2.15 (1H, m), 2.24 (1H, m), 2.77 (3H, s), 2.78 (3H, s), 2.94 (3H, s), 3.10 (1H, s), 3.69 (3H, s), 3.87 (1H, br), 4.10 (1H, br), 4.41 (1H, m), 4.50 (1H, ddd, $J$ = 5.7, 8.1, 13.8 Hz), 4.72 (1H, dd, $J$ = 5.7, 8.0 Hz), 4.91–4.98 (3H, m), 5.03 (1H, dd, $J$ = 6.5, 9.1 Hz), 5.18 (1H, m), 5.48 (1H, m), 5.62 (1H, m), 6.53 (1H, d, $J$ = 7.9 Hz), 6.92 (1H, d, $J$ = 8.5 Hz), 7.60 (1H, d, $J$ = 7.5 Hz). [13]C NMR (150 MHz, CDCl$_3$): major conformer δ 11.9, 14.1, 14.2, 14.6, 17.2, 19.7, 19.8, 20.0, 21.9, 23.2, 24.8, 25.7, 26.5, 28.4 (3C), 29.0, 29.8, 30.3, 30.5, 30.7, 34.5, 36.2, 36.8, 38.7, 50.4, 50.5, 50.6, 50.7, 52.4, 52.7, 52.8, 54.3, 55.0, 79.1, 113.9, 142.8, 155.9, 170.1, 171.0, 171.3, 171.7, 172.1, 172.2, 172.7. LC-MS (ESI +): 908.52 (+).

## (S)-Boc-photo-leucine methyl ester (16)

To a solution of (S)-Boc-photo-leucine (72 mg, 0.3 mmol) in MeOH (4 mL) at RT was added TMS-diazomethane (2 M in hexane) dropwise until a yellow color persisted. The reaction was then concentrated, giving **16** as a clear oil (85 mg, quantitative yield). [1]H NMR (400 MHz, CDCl$_3$): δ 1.05 (3H, s), 1.44 (9H, s), 1.86 (1H, m), 3.73 (3H, s), 4.34 (1H, m), 5.12 (1H, m). [13]C NMR (100 MHz, CDCl$_3$): δ 19.8, 23.8, 28.4, 38.2, 50.3, 52.7, 80.3, 155.1, 172.3. LC-MS (ESI –): 256.14 (–).

**Figure 13.** Compound 18.

   [1]H NMR (500 MHz, CDCl$_3$): major conformer δ 0.84–1.05 (24H, m), 1.28–1.40 (8H, m), 1.44 (9H, s), 1.61–1.77 (5H, m), 1.94–2.06 (4H, m), 2.13 (1H, dd, $J$ = 4.3, 15 Hz), 2.82 (3H, s), 2.83 (3H, s), 2.95 (3H, s), 3.12 (1H, s), 3.74 (3H, s), 4.10 (1H, d, $J$ = 6.3 Hz), 4.40–4.36 (2H, m), 4.61 (1H, td, $J$ = 8.9, 4.0 Hz), 4.75 (1H, dt, $J$ = 6.6, 3.4 Hz), 4.84–4.86 (1H, m), 5.01–5.03 (1H, m), 5.09–5.12 (1H, m), 5.14–5.18 (1H, m), 5.63 (1H, q, $J$ = 6.8 Hz), 6.69 (1H, d, $J$ = 8.5 Hz), 6.84 (1H, d, $J$ = 8.1 Hz), 7.07 (1H, d, $J$ = 8.2 Hz).[13]C NMR (125 MHz, CDCl$_3$): major conformer δ 11.7, 11.8, 14.0, 14.2, 14.5, 17.9, 19.4, 19.6, 22.8 (3C), 23.3, 23.8, 24.7, 24.8, 26.6, 28.4, 30.3, 30.4, 35.7, 37.3, 48.7, 49.4, 50.7, 52.7, 52.8, 52.9, 54.6, 54.7, 56.2, 62.8, 70.6, 70.7, 75.7, 79.6, 80.4, 80.9, 156.0, 169.2, 169.9, 170.0, 171.3, 171.4, 172.6, 173.9. LC-MS (ESI –): 904.74 (–).

**Figure 14.** Compound 3.

Using the general procedures described above, the fully protected linear heptapeptide **13** (60 mg, 0.066 mmol) was deprotected and converted to cyclic peptide **3** (12 mg, 24%). [1]H NMR (600 MHz, DMSO-*d6*): δ 0.79–0.95 (21H, m), 1.05 (3H, d, *J* = 6.4 Hz), 1.21 (5H, s), 1.26 (3H, d, *J* = 7.2 Hz), 1.34–1.42 (5H, m), 1.50 (4H, dd, *J* = 21.3, 8.1 Hz), 1.61 (3H, t, *J* = 15.6 Hz), 1.76–1.80 (3H, m), 1.89–1.98 (2H, m), 2.35 (1H, d, *J* = 12.7 Hz), 2.47 (2H, dt, *J* = 3.6, 1.8 Hz), 2.79 (3H, s), 2.84 (3H, s), 3.02 (3H, s), 3.42 (1H, d, *J* = 9.4 Hz), 3.88 (1H, dd, *J* = 10.9, 4.0 Hz), 4.04 (1H, d, *J* = 12.9 Hz), 4.47 (1H, t, *J* = 4.3 Hz), 4.78 (2H, d, *J* = 32.4 Hz), 4.86 (1H, d, *J* = 1.5 Hz), 4.97 (2H, d, *J* = 27.5 Hz), 5.31 (1H, q, *J* = 6.5 Hz), 7.41 (1H, d, *J* = 9.8 Hz), 7.74 (1H, d, *J* = 9.0 Hz), 8.62 (1H, d, *J* = 4.9 Hz).[13]C NMR(150 MHz, DMSO-*d6*): δ 11.7, 13.5, 13.7, 14.6, 15.3, 20.6, 20.8, 20.9, 22.0, 23.1, 23.5, 24.4, 25.1, 25.2, 25.3, 25.7, 28.4, 29.1, 29.4, 31.1, 33.3, 37.5, 43.1, 43.0, 48.7, 49.2, 50.6, 51.8, 54.4, 55.0, 57.9, 75.3, 167.6, 168.0, 168.6, 172.0, 172.6, 172.7, 173.5. LC-MS (ESI –): 762.51 (–).

**Figure 15.** Compound 4.

Using the general procedures described above, the fully protected linear heptapeptide **14** (78 mg, 0.078 mmol) was deprotected and converted to cyclic peptide **4** (8.7 mg, 22%). [1]H NMR (600 MHz, DMSO-*d6*): δ 0.82 (3H, d, *J* = 6.7 Hz), 0.83–0.96 (20H, m), 1.09 (3H, d, *J* = 6.5 Hz), 1.23 (2H, s), 1.28 (3H, d, *J* = 7.4 Hz), 1.38 (2H, m), 1.52 (2H, m), 1.58–1.67 (3H, m), 1.80 (2H, m), 1.94 (1H, m), 2.05 (1H, m), 2.38 (1H, m), 2.83 (3H, s), 2.90 (1H, m), 2.89 (3H, s), 2.99 (3H, s), 3.44 (1H, m), 3.90 (1H, dd, *J* = 4.1, 10.9 Hz), 4.07 (1H, d, *J* = 12.6 Hz), 4.52 (1H, dd, *J* = 3.9, 4.7 Hz), 4.67 (1H, m), 4.81 (1H, t, *J* = 9.6 Hz), 4.88 (2H, m), 5.00 (2H, m), 5.34 (1H, dd, *J* = 6.5, 13 Hz), 5.69 (1H, ddd, *J* = 8.7, 10.0, 17.0 Hz), 7.39 (1H, d, *J* = 9.8 Hz), 7.76 (1H, d, *J* = 5.3 Hz), 8.69 (1H, d, *J* = 4.9 Hz). [13]C NMR (150 MHz, DMSO-*d6*): δ 11.7, 13.6, 13.7, 14.6, 15.4, 20.7, 20.8, 22.1, 23.6, 24.5, 25.2, 25.3, 25.7, 28.4, 29.1, 29.5, 31.1, 33.4, 35.5, 35.6, 35.5, 35.7, 43.2, 48.7, 49.4, 50.7, 51.9, 54.4, 55.1, 58.1, 75.4, 115.1, 142.5, 167.8, 168.1, 168.7, 172.2, 172.7 (2C), 173.7. LC-MS (ESI +): 776.57 (+).

**Figure 16.** Compound 4b.

Using the general procedures described above, the fully protected linear heptapeptide **15** (50 mg, 0.065 mmol) was deprotected and converted to cyclic peptide **4b** (5.8 mg, 12%). [1]H NMR (600 MHz, DMSO-*d6*): δ 0.82 (3H, d, *J* = 6.7 Hz), 0.84–0.90 (15H, m), 0.93 (3H, d, *J* = 6.9 Hz), 0.98 (3H, d, *J* = 6.7 Hz), 1.09 (3H, d, *J* = 6.5 Hz), 1.23 (1H, m), 1.29 (3H, d, *J* = 7.4 Hz), 1.39 (2H, m), 1.50 (2H, m), 1.64 (3H, m), 1.80 (2H, m), 1.94 (1H, ddd, *J* = 4.6, 11.0, 13.1 Hz), 2.14 (1H, m), 2.37 (1H, m), 2.83 (3H, s), 2.88 (3H, s), 2.90 (1H, m) 3.06 (3H, s), 3.45 (1H, dd, *J* = 1.8, 9.5 Hz), 3.91 (1H, dd, *J* = 4.2, 10.7 Hz), 4.06 (1H, d, *J* = 12.8 Hz), 4.49 (1H, dd, *J* = 3.7, 4.9 Hz), 4.80 (1H, t, *J* = 9.6 Hz), 4.84 (1H, m), 4.86 (1H, m), 4.85–5.04 (3H, m), 5.33 (1H, dd, *J* = 6.2, 12.8 Hz), 5.74 (1H, app sept., *J* = 6.9, 10.3, 17.2 Hz),7.40 (1H, d, *J* = 9.8 Hz), 7.81 (1H, d, *J* = 8.9), 8.62 (1H, d, *J* = 4.9 Hz). [13]C NMR (125 MHz, DMSO-*d6*): δ 11.6, 13.5, 13.7, 14.6, 15.3, 17.8, 20.6, 20.8, 22.0, 22.5, 23.5, 24.4, 25.1, 25.2, 25.7, 28.4, 29.1, 29.5, 31.2, 33.1, 34.2, 35.2, 43.1, 48.4, 49.2, 50.7, 51.8, 54.4, 54.9, 57.9, 75.2, 112.6, 143.8, 167.7, 168.0, 168.6, 171.9, 172.6, 172.7, 173.5. LC-MS (ESI +): 776.52 (+).

**5**

**Figure 17.** Compound 5.

Using the general procedures described above, the fully protected linear heptapeptide **17** (45 mg, 0.044 mmol) was deprotected and converted to cyclic peptide **5** (9.2 mg, 27%). [1]H NMR (600 MHz, DMSO-*d6*): δ 0.81 (3H, d, *J* = 6.6 Hz), 0.84-0.92 (10H, m), 1.01 (3H, s), 1.11 (3H, d, *J* =6.5 Hz), 1.13-1.41 (11H, m), 1.47-1.55 (2H, m), 1.61-1.71 (3H, m), 1.82-1.86 (1H, m), 1.90 (1H, dd, *J* = 2.1, 15.5 Hz), 2.00 (1H, ddd, *J* = 5.8, 12.2, 17.9 Hz), 2.55-2.60 (1H, m), 2.74-2.80 (2H, m), 2.89 (3H, s), 2.91 (3H, s), 3.02 (3H, s), 3.04 (3H, s), 3.37 (1H, d, *J* = 9.3 Hz),3.98 (1H, dd, *J* = 3.8, 11.1 Hz), 4.12-4.14 (1H, m), 4.61 (1H, dd, *J* = 4.8, 3.6 Hz), 4.70 (1H, t, *J* = 9.6 Hz), 4.77 (1H, ddd, *J* = 11.7, 9.2, 2.3 Hz), 4.86 (1H, q, 7.2 Hz), 5.19 (1H, dd, *J* = 11.4, 4.7 Hz), 5.39 (1H, q, *J* = 6.6 Hz), 7.15 (1H, d, *J* = 9.8 Hz), 7.1 (1H, m), 8.55 (1H, d, *J* = 5.0 Hz). [13]C NMR (150 MHz, C$_6$D$_6$): δ 11.7, 13.5, 14.4, 14.7, 19.4, 21.3, 22.5, 24.2, 25.4, 26.8, 29.3, 29.4, 29.9, 30.2, 30.2, 30.3, 30.5, 30.9, 33.5, 34.2, 39.2, 49.8, 51.7, 55.3, 55.3, 56.3, 59.2, 68.1, 70.1, 75.9, 80.5, 167.6, 168.1, 169.0, 172.5, 173.1, 174.3, 175.4. LC-MS (ESI –): 772.69 (–).

## Acknowledgements

Funding for this study was provided by the Howard Hughes Medical Institute (JT). We thank Helen Pham for facilitating compound handling, Dominic Hoepfner for providing mutant and parental HCT116 cell lines, and Milan Cvitkovic for synthesizing hydroxy-leucine.

## Additional information

### Funding

| Funder | Author |
| --- | --- |
| Howard Hughes Medical Institute | Jack Taunton |

The funders had no role in study design, data collection and interpretation, or the decision to submit the work for publication.

### Author contributions

JDC, Conception and design, Acquisition of data, Analysis and interpretation of data, Drafting or revising the article; SGS, Conception and design, Acquisition of data, Drafting or revising the article;

GAS, HRM, JLS, Acquisition of data, Drafting or revising the article; WCM, Analysis and interpretation of data, Drafting or revising the article; RKJ, NTR, Analysis and interpretation of data, Drafting or revising the article, Contributed unpublished essential data or reagents; JT, Conception and design, Analysis and interpretation of data, Drafting or revising the article

## Author ORCIDs
Jordan D Carelli, http://orcid.org/0000-0001-7625-7505

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
