## [Decision Letter]

Thank you for submitting your work entitled "Ternatin and improved synthetic variants kill cancer cells by targeting the elongation factor-1A ternary complex" for peer review at *eLife*. Your submission has been favorably evaluated by John Kuriyan (Senior editor), a Reviewing editor, and three reviewers.

Summary:

*Reviewer 1:* This is a fascinating detective story epitomizing the power of combining chemical tools with biological intuition to elucidate the mode of action of a promising family of natural products and related analogs. The strengths of the manuscript include (1) Synthesis and discovery of a new derivative, ternatinA4, with significantly improved cellular activity over ternatin itself. (2) Demonstration, once again, of the use of photo-leucine, as a useful tool to help detect binding proteins of cyclic peptides. (3) Elucidation of the mode of action of the ternatin family of natural products and identification of the eEF1A-GTP-aminoacyl-tRNA complex as the target. Though ternatin and its analog ternatinA4 are not the first translation inhibitors targeting eEF1A, their structural simplicity and ease of synthesis renders them more ideal leads for future drug development. (4) The data are of high quality and convincing.

*Reviewer 2:* The study nicely demonstrates the utility of chemical biology approaches for target identification and has identified what appears to be the major cellular target of a class of compounds with therapeutic potential. Overall, it is a nice body of work that will be of interest to scientists in the fields of chemical biology and drug discovery.

*Reviewer 3:* In this manuscript by Carelli and co-workers the authors seek to identify the macromolecular target of the ternatin natural product and related derivatives. The authors use similarities between ternatin and other reported cytotoxic natural products to design hybrid compounds (3 and 4), and these molecules are more potent as cell death inducing agents. After showing these compounds inhibit protein synthesis, the authors use a series of labeling studies (utilizing derivative 5) to infer that eEF1A is the biological target. Interestingly, the authors never apparently confirm this target with mass spec (as is standard nowadays) but instead use resistant cell lines that have mutations in this target, and some in vitro experiments, to connect the compound to the target. Overall this type of target ID work is messy and difficult and the authors are commended for doing a very nice job with this study.

Major comments for revision:

1) *Reviewer 2:* The authors use ternatin-resistant cell lines containing A399V/T mutations, however these lines appear to have been generated by selection against another protein synthesis inhibitor. In light of this, these cell lines may contain additional mutations that are necessary for ternatin resistance. I suggest that the authors demonstrate that overexpression of a mutant eEFA1 transgene confers similar cellular resistance to the drug.

2) *Reviewer 3:* In my opinion, it is a missed opportunity that the authors do not come back and try to explain the results of the 21 cell line panel in the context of their data. Could it be that the three highly resistant cell lines have these very mutations in the EE1A1 gene (that confer resistance to 4, Figure 5)? That would be incredibly exciting. And, actionable – as the authors mention this eEF1A is not an uncommon target for drugs to hit, and experimental therapeutics that hit this target are in the clinic. If patient outcome could be predicted before treatment (based on this mutation) that would be a most valuable part of this work.

Full reviews:

Reviewer #1:

In this manuscript, Taunton and colleagues elucidated the molecular mechanism of action of ternatin and its synthetic analog with greater antiproliferative potency. The authors began with an astute detective work to identify the relationship between ternatin and some of highly potent antiproliferative cyclic peptides disclosed in the patent literature. They then synthesized a few analogs and identified one (ternatin analog #4 or ternatinA4) with 20- to 50-fold higher potency than the ternatin itself in a large number of cancer cell lines. Based on limited structure-activity relationship, an active photo-affinity label with a clickable handle was made and used to detect a 50-kD protein from cell extract. Guided by the effect of ternatin and analogs on translation elongation, the authors guessed that the 50-kD protein is eEF1A, a GTPase that is involved in the loading of charged tRNA during the elongation phase of protein synthesis. Subsequently, the authors found that ternatinA4 indeed bound to eEF1A in a GTP and aminoacyl-tRNA dependent manner, suggesting that ternatin bound to the ternary complex of eEF1A, GTP and aminoacyl-tRNA. To confirm that eEF1A is the bona fide target for ternatinA4, the authors took advantage of a known Ala399 mutant of eEF1A that was recently reported to confer resistance to another elongation inhibitor nannocystin A. Indeed, the same A399V mutant was found to confer resistance to ternatinA4 to cells and lost binding to ternatinA4. Moreover, the authors also demonstrated that the known eEF1A inhibitors didemnin B and ansatrienin B could both compete against the ternatin photo-affinity label for binding to eEF1A, suggesting that they share the same target.

This is a fascinating detective story epitomizing the power of combining chemical tools with biological intuition to elucidate the mode of action of a promising family of natural products and related analogs. The strengths of the manuscript include (1) Synthesis and discovery of a new derivative, ternatinA4, with significantly improved cellular activity over ternatin itself. (2) Demonstration, once again, of the use of photo-leucine, as a useful tool to help detect binding proteins of cyclic peptides. (3) Elucidation of the mode of action of the ternatin family of natural products and identification of the eEF1A-GTP-aminoacyl-tRNA complex as the target. Though ternatin and its analog ternatinA4 are not the first translation inhibitors targeting eEF1A, their structural simplicity and ease of synthesis renders them more ideal leads for future drug development. (4) The data are of high quality and convincing.

A few issues remain that the authors may consider addressing to further strengthen the manuscript.

1) Very minor: A gradient symbol was used to show changes in concentrations of compounds used in different experiments (i.e., Figure 2, Figure 3, Figure 4). Though the lowest and highest concentrations were shown, it is not immediately obvious what those intermediate concentrations were. As there were only a limited number of lanes, the drug concentration for each lane should be either shown in the figure or figure legends.

2) Minor: In Figure 2, the authors used inhibition of polysome depletion by an initiation inhibitor as a hint that ternatinA4 likely inhibits the elongation phase of protein synthesis. A more definitive and popular method is to use a dual luciferase reporter with one of the luciferases under the control of an IRES element from a viral transcript. This can be easily done and will complement the polysome depletion assay.

3) Major but optional: Since it has been shown that ternatinA4 binds to the eEF1A-GTP-aminoacyl-tRNA complex, an obvious mechanistic question is what is the functional impact of this interaction on the complex. A couple of obvious questions that come to mind include 1) Does ternatinA4 affect the binding of the eEF1A-GTP-aminoacyl-tRNA complex to the ribosome? 2) Does ternatinA4 inhibit the ribosome-stimulated GTP hydrolysis of the eEF1A complex? It is noted that the authors may not have the assays established in the lab. Hence, these experiments are optional.

Reviewer #2:

The authors describe their efforts to elucidate the mechanism of action of the cytotoxic natural product, ternatin, as well as related compounds. Guided by a photocrosslinkable and clickable ternatin derivative that they design, and cell biological investigation of the effects of the compound on protein synthesis, they identify eEFA1 as the cellular target of ternatin. Their assertion is supported by the demonstration that a single point mutation in eEFA1 abolishes photocrosslinking to their ternatin probe and confers cellular resistance to ternatin.

The study nicely demonstrates the utility of chemical biology approaches for target identification and has identified what appears to be the major cellular target of a class of compounds with therapeutic potential. Overall, it is a nice body of work that will be of interest to scientists in the fields of chemical biology and drug discovery.

I have only two concerns about the manuscript in its present state:

1) The authors claim that ternatin binds to a ternary complex containing GTP and aminoacyl-tRNA. Their claim is supported by RNaseA digestion of lysate that abrogates cross-linking to the ternatin probe and their reconstitution of the ternary complex. My concern relates to the resconstitution experiment that relies upon Phe-tRNA obtained from extracts. It seems that in this context the authors cannot exclude that some other RNA component may be important for binding, or that eEFA1 is somehow processed by residual activity carried over from the extract. There is no description of the purification protocol used to isolate the tRNA. Furthermore, why was Phe-tRNA used? Will any tRNA suffice or is ternatin binding specific to a complex containing Phe-tRNA?

2) The authors use ternatin-resistant cell lines containing A399V/T mutations, however these lines appear to have been generated by selection against another protein synthesis inhibitor. In light of this, these cell lines may contain additional mutations that are necessary for ternatin resistance. I suggest that the authors demonstrate that overexpression of a mutant eEFA1 transgene confers similar cellular resistance to the drug.

If these concerns can be addressed, then this manuscript should be published in *eLife*.

Reviewer #3:

In this manuscript by Carelli and co-workers the authors seek to identify the macromolecular target of the ternatin natural product and related derivatives. The authors use similarities between ternatin and other reported cytotoxic natural products to design hybrid compounds (3 and 4), and these molecules are more potent as cell death inducing agents. After showing these compounds inhibit protein synthesis, the authors use a series of labeling studies (utilizing derivative 5) to infer that eEF1A is the biological target. Interestingly, the authors never apparently confirm this target with mass spec (as is standard nowadays) but instead use resistant cell lines that have mutations in this target, and some in vitro experiments, to connect the compound to the target.

Overall this type of target ID work is messy and difficult and the authors are commended for doing a very nice job with this study.

There are a few issues/questions, some of which could make the paper much more interesting and useful if answered:

1) The authors assess compounds 1 and 4 in a 21 cell line panel, but this panel does not include the workhorse cell line for this study (HEK293T), very odd. Where does this cell line fall in the two orders of magnitude sensitivity spectrum? That information would be useful in interpreting if the results (Figure 3 and Figure 4) are expected to be typical, or are anomalous.

2) Perhaps more importantly, in my opinion it is a missed opportunity that the authors do not come back and try to explain the results of the 21 cell line panel in the context of their data. Could it be that the three highly resistant cell lines have these very mutations in the EE1A1 gene (that confer resistance to 4, Figure 5)? That would be incredibly exciting. And, actionable – as the authors mention this eEF1A is not an uncommon target for drugs to hit, and experimental therapeutics that hit this target are in the clinic. If patient outcome could be predicted before treatment (based on this mutation) that would be a most valuable part of this work.

3) Again, it is strange that the authors go to the trouble to generate nannocystin-resistant HCT116 cells – why not generate cells that are directly resistant to 1 or 4? I assume this was attempted.

4) There are many places where key data is mentioned but not shown anywhere (effect on actin/tubulin, the nannocystin-resistance of the cell lines, etc.). Most journals now have a "no data not shown" policy.

5) "% DMSO control" or "Fraction DMSO control" (in SI) are used for the IC50 curve Y-axes. More standard would be "%AV/PI positive" or "% cell death"; the figure legend should state how long the compound/cell incubation is in these experiments, as that can influence the interpretation of the data.

---

## [Author Response]

*Major comments for revision:1)* Reviewer 2: *The authors use ternatin-resistant cell lines containing A399V/T mutations, however these lines appear to have been generated by selection against another protein synthesis inhibitor. In light of this, these cell lines may contain additional mutations that are necessary for ternatin resistance. I suggest that the authors demonstrate that overexpression of a mutant eEFA1 transgene confers similar cellular resistance to the drug.*

Because resistance to ternatin is recessive (heterozygous A399 mutants are only partially resistant), and endogenous eEF1A is one of the most abundant proteins in the cell, the proposed experiment is challenging. Despite many attempts with various expression vectors, we were unable to express mutant eEF1A at high enough levels to confer significant resistance in a 'wild-type' cell line.

We therefore devised an alternative experiment, the results of which strongly support our central conclusion that eEF1A is the relevant target of ternatin cyclic peptides. Starting with homozygous A399V EEF1A1 cells, which are completely resistant to compound 4, we show that ectopic expression of WT eEF1A is sufficient to restore sensitivity. By contrast, A399V cells transduced with A399V eEF1A remain completely resistant. These new data are included in Figure 5.

*2)* Reviewer 3: *In my opinion, it is a missed opportunity that the authors do not come back and try to explain the results of the 21 cell line panel in the context of their data. Could it be that the three highly resistant cell lines have these very mutations in the EE1A1 gene (that confer resistance to 4, Figure 5)? That would be incredibly exciting. And, actionable – as the authors mention this eEF1A is not an uncommon target for drugs to hit, and experimental therapeutics that hit this target are in the clinic. If patient outcome could be predicted before treatment (based on this mutation) that would be a most valuable part of this work.*

We appreciate the reviewer’s enthusiasm and agree this is an important question. We note that mechanisms underlying intrinsic resistance of cancer cells to cytotoxic drugs are often complex and multifactorial. Nevertheless, to address the reviewer's intriguing question, we scrutinized internal RNAseq data (available for 18 of the 21 cell lines). We did not identify any EEF1A1 or EEF1A2 mutations at A399. Mutations at other EEF1A1 positions were tentatively identified in two of the three most resistant cell lines, as well as one highly sensitive cell line. However, the functional relevance of these mutations is unknown and currently under investigation.

Full reviews:

Reviewer #1:

[…] A few issues remain that the authors may consider addressing to further strengthen the manuscript.

1) Very minor: A gradient symbol was used to show changes in concentrations of compounds used in different experiments (i.e., Figure 2, Figure 3., Figure 4). Though the lowest and highest concentrations were shown, it is not immediately obvious what those intermediate concentrations were. As there were only a limited number of lanes, the drug concentration for each lane should be either shown in the figure or figure legends.

We have adjusted the figure legends to improve clarity.

2) Minor: In Figure 2, the authors used inhibition of polysome depletion by an initiation inhibitor as a hint that ternatinA4 likely inhibits the elongation phase of protein synthesis. A more definitive and popular method is to use a dual luciferase reporter with one of the luciferases under the control of an IRES element from a viral transcript. This can be easily done and will complement the polysome depletion assay.

We agree this is an interesting experiment that would complement the polysome depletion assay. However, we have chosen to focus our efforts on generating additional genetic data that implicate eEF1A as the biologically relevant target of ternatin-family cyclic peptides. Experiments to define the precise mechanism by which ternatins inhibit protein synthesis will form the basis of a future study.

3) Major but optional: Since it has been shown that ternatinA4 binds to the eEF1A-GTP-aminoacyl-tRNA complex, an obvious mechanistic question is what is the functional impact of this interaction on the complex. A couple of obvious questions that come to mind include 1) Does ternatinA4 affect the binding of the eEF1A-GTP-aminoacyl-tRNA complex to the ribosome? 2) Does ternatinA4 inhibit the ribosome-stimulated GTP hydrolysis of the eEF1A complex? It is noted that the authors may not have the assays established in the lab. Hence, these experiments are optional.

These are exciting mechanistic questions that we hope to address in future studies.

Reviewer #2:

[…] I have only two concerns about the manuscript in its present state:

1) The authors claim that ternatin binds to a ternary complex containing GTP and aminoacyl-tRNA. Their claim is supported by RNaseA digestion of lysate that abrogates cross-linking to the ternatin probe and their reconstitution of the ternary complex. My concern relates to the resconstitution experiment that relies upon Phe-tRNA obtained from extracts. It seems that in this context the authors cannot exclude that some other RNA component may be important for binding, or that eEFA1 is somehow processed by residual activity carried over from the extract. There is no description of the purification protocol used to isolate the tRNA. Furthermore, why was Phe-tRNA used? Will any tRNA suffice or is ternatin binding specific to a complex containing Phe-tRNA?

While we cannot completely exclude the presence of trace RNA contaminants in our photo-affinity labeling experiments, it is important to note that we did not use extracts to generate Phe-tRNA. Bulk tRNA was purified from reticulocytes according to Merrick (1979) and is free of mRNA, 18S rRNA, and 28S rRNA. After charging bulk tRNA with Phe using a partially purified aminoacyl tRNA synthetase fraction (purified by DEAE chromatography according to Merrick, 1979), Phe-tRNA was isolated by sequential phenol-chloroform extraction, EtOH precipitation, and Sephadex G-25 chromatography (Walker and Fredrick, 2008). Phe-tRNA is often used in biochemical reconstitution experiments due to its relative stability. Whether ternatin binds eEF1A in the presence other aa-tRNAs is an intriguing question that merits further investigation.

Reviewer #3:

[…] There are a few issues/questions, some of which could make the paper much more interesting and useful if answered:

1) The authors assess compounds 1 and 4 in a 21-cell line panel, but this panel does not include the workhorse cell line for this study (HEK293T), very odd. Where does this cell line fall in the two orders of magnitude sensitivity spectrum? That information would be useful in interpreting if the results (Figure 3 and Figure 4) are expected to be typical, or are anomalous.

We have added proliferation data for HEK293T cells (Figure 3—figure supplement 1), which show essentially the same sensitivity as HCT116 cells.

3) Again, it is strange that the authors go to the trouble to generate nannocystin-resistant HCT116 cells – why not generate cells that are directly resistant to 1 or 4? I assume this was attempted.

The nannocystin-resistant, *EEF1A1*-mutant cell lines were generated in a separate study (Krastel et al., 2015); we obtained and tested these cells because our experiments with ternatin implicated eEF1A as the likely target.

4) There are many places where key data is mentioned but not shown anywhere (effect on actin/tubulin, the nannocystin-resistance of the cell lines, etc.). Most journals now have a "no data not shown" policy.

As mentioned above, the nannocystin-resistant cell lines were published in a separate study (Krastel et al., 2015). We have removed the sentence that refers to a negative result regarding effects on actin/tubulin.

*5) "% DMSO control" or "Fraction DMSO control" (in SI) are used for the IC50 curve Y-axes. More standard would be "%AV/PI positive" or "% cell death"; the figure legend should state how long the compound/cell incubation is in these experiments, as that can influence the interpretation of the data.*

We have made the requested changes.